# Uterine prolapse and associated factors among reproductive-age women in south-west Ethiopia: A community-based cross-sectional study

Abebe Sorsa Badacho๏*, Mengistu Auro Lelu, Zegeye Gelan, Deginesh Dawit Woltamo

School of Public Health, College of Health Sciences and Medicine, Wolaita Sodo University, Wolaita Sodo, Ethiopia

* abebe.sorsa@wsu.edu.et, sorsabebe@gmail.com

## Abstract

### Background

Uterine prolapse is an important but neglected public health problem that causes maternal morbidity and mortality in women of reproductive age in low- and middle-income countries, including Ethiopia. However, little data are available concerning uterine prolapse in Ethiopia. The objective of this study was to assess the prevalence of and factors associated with uterine prolapse in women of reproductive age in Ethiopia.

### Methods

A community-based cross-sectional study was conducted in Loma Woreda, Dawro, south-west Ethiopia, in November and December 2019. Four hundred and twenty-two randomly selected women of reproductive age participated in the study. Face-to-face interviews with a pre-structured questionnaire collected data, and diagnoses were made clinically. Epi Data v3.2.1 and SPSS v24 were used for data entry, processing, and analysis. Binary logistic regression was used to determine associations between dependent and independent variables. Variables with P-values less than 0.25 in bivariate logistic regression were further examined using multivariate logistic regression to investigate associations between the dependent variable and independent variables.

### Results

The mean age of respondents was 35.4 ±7.994 years. The prevalence of symptomatic and anatomical uterine prolapse was 6.6% (28) and 5.9% (25), respectively. The prevalence of anatomical prolapse was used as a reference when determining associated factors. Age at first marriage (Adjusted Odd Ratio (AOR): 0.25, 95%CI 0.07, 0.89), place of delivery (AOR: 3.33, 95%CI 1.21, 9.13), birth attendant-assisted delivery (AOR 0.21; 95%CI 0.06, 0.71), and history of abortion (AOR: 2.94, 95%CI 1.08, 7.97) were found significantly and independently associated with the prevalence of uterine prolapse.

**Data Availability Statement:** All relevant data are within the manuscript and its Supporting Information files.

**Funding:** Unfunded studies: The author(s) received no specific funding for this work.

**Competing interests:** NO authors have competing interests: The authors have declared that no competing interests exist

**Abbreviations:** ANC, Antenatal Care; BMI, Body Mass Index; POP, Pelvic Organ Prolapse; UK, United Kingdom; UP, Uterine Prolapse; US, United States; USA, United States of America; UVP, Uterine Vaginal Prolapse; VAGH, Vaginal Hysterectomy.

## Conclusion

Uterine prolapse is common in women of reproductive age. Age at first marriage, place of delivery, birth attendant-assisted delivery, and history of abortion were independent predictors of the prevalence of uterine prolapse. We recommend that the health system link primary health care to hospital-set for uterine prolapse treatment programs. Health institution delivery should be encouraged by the local government. Early marriage and unwanted pregnancy need to be prevented through appropriate strategies.

## Introduction

Uterine prolapse (UP), also known as pelvic organ prolapse (POP) and genital prolapse, describes the descent of the uterus from its normal anatomical confines to positions within or outside the vaginal introitus. UP occurs secondary to weakened pelvic muscles that can no longer support the appropriate positioning of the pelvic organs and can be accompanied by different prolapse symptoms like a feeling of heaviness and sexual, urinary, and bowel dysfunction [1].

UP is the most common gynecological health problem contributing to maternal morbidity and mortality in women of reproductive age in developing countries. It leads to varying degrees of physical disability, including an inability to work, difficulties in walking or standing up, difficulties in urinating or defecating, painful intercourse, increased social stigma, and economic deprivation. UP can also affect women's mental health and can be fatal if left untreated [1–3]. Due to stigma in low-income countries, women affected by UP often hide their condition, do not seek help, and live with the disease and its complications for long periods [4].

The worldwide prevalence of UP has been reported to be around 9%. However, in low and middle-income countries (LMICs), it is estimated to be nearly 20%, and estimates vary widely (3.4–56.4%) [5]. The prevalence based on symptoms is 3–6% and up to 50% when defined by vaginal checkups [6].

The burden of UP in low-income countries is expected to be worse than that of developed countries, given the low level of awareness of women in developing countries [4, 7, 8].

Major risk factors associated with UP are adolescent pregnancy, lack of rest during and immediately after pregnancy, carrying heavy loads, delivery by unskilled birth attendants, poor nutrition, frequent pregnancies and pregnancies close together, prolonged and obstructed labor, and weakening of pelvic muscles as a result of aging or other medical problems [3, 9].

UP is, therefore, an important but one of the most neglected public health problems in LMICs, including Ethiopia, where there is little literature regarding its prevalence [10, 11]. Indeed, there have been no published population-based studies on UP in Ethiopia, although reports from individual hospitals suggest a high burden of UP among women at gynecological outpatient clinics and wards [5, 9].

The high numbers of women potentially affected by UP and the paucity of locally generated evidence on the magnitude and associated factors of UP to design appropriate prevention strategies [10, 12], here we assessed the prevalence of and factors associated with uterine prolapse in women of reproductive age in Ethiopia (Fig 1).

## Materials and methods

### Study design and period

A community based cross-sectional study was conducted in November and December 2019.

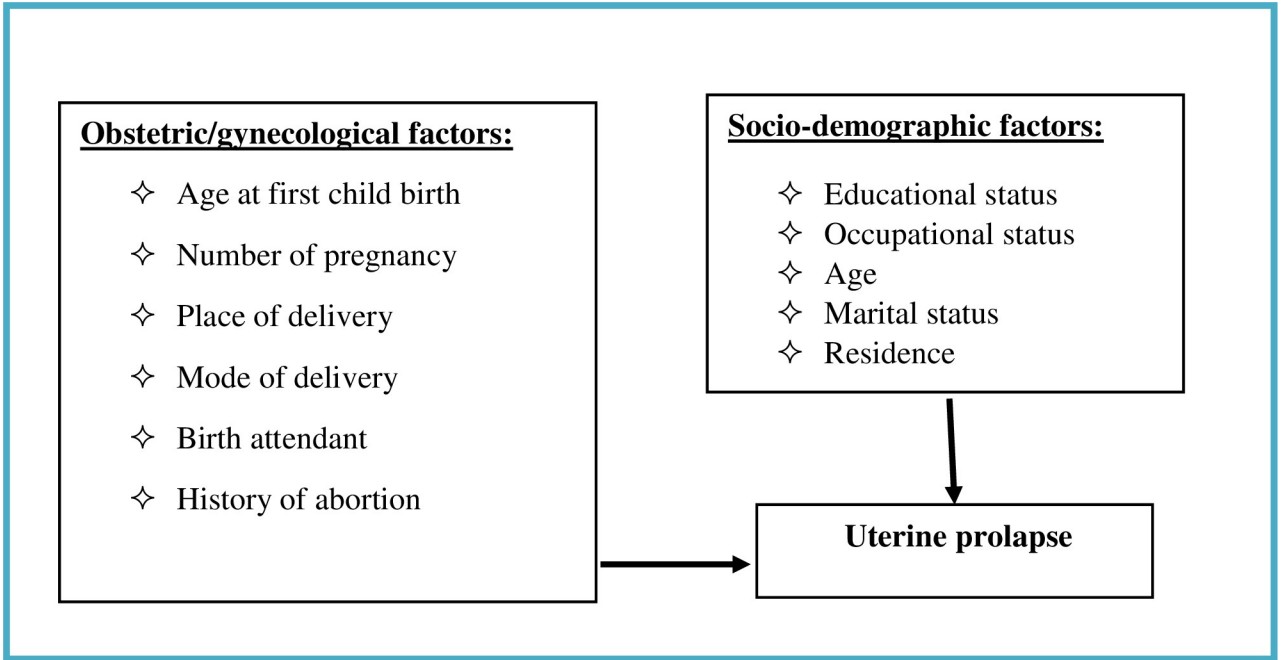

**Fig 1. Conceptual framework of the study developed from reviewing the literature.**

### Source and study population

The source or target population of the study was all women of the reproductive age group who had a history of at least one delivery, while the study population was all women randomly selected.

### Inclusion criteria

All ever-married and single women of reproductive age group greater and equal 18 years who had a history of at least one delivery were included. Pregnant women were excluded.

### Sample size determination and sampling procedure

The sample size was calculated to ensure that the two-sided 95% confidence interval (CI) for the estimated prevalence will be within +/- 0.05 by using a single population formula with a proportion of 0.5.

Four hundred twenty-two randomly selected women were involved in the study using a household as a sampling frame. From the total 28 kebeles in the Loma district, 30% of kebeles, i.e., eight kebeles, which is the lowest administrative unit in Ethiopia, were selected by the lottery method. We raffled eight areas of the Loma district and then selected the sample by a simple random sampling in these areas. The assumption was to divide the total estimated sample size to the households of each kebele according to the proportion they contribute to the total study subjects. We allocated sample proportion for the selected eight kebeles based on sample size. Out of an estimated 422 participants, the sample size was adjusted proportionally for the selected eight kebeles. Then, married women of reproductive age were selected by random sampling method, using a married reproductive age group list from a registration book of health post as provided by Health Extension Workers (HEWs) working at Health Post in each respective kebele." Table 1.

**Table 1. Proportional allocation of study participants from each kebele in the Loma woreda, Dawuro Zone, Ethiopia 2020.**

| Kebeles | Kebele 1 | Kebele 2 | Kebele 3 | Kebele 4 | Kebele 5 | Kebele 6 | Kebele7 | Kebele 8 |
|---|---|---|---|---|---|---|---|---|
| Total number of married Reproductive age group in households | 673 | 735 | 812 | 741 | 620 | 65 | 705 | 720 |
| Number of women participated in the study | 50 | 55 | 61 | 55 | 46 | 48 | 53 | 54 |

## Data collection

Eight trained midwives collected data through face-to-face interviews at participants' homes supervised by two BSc-qualified Nurse Supervisors.

## Quality assurance

Data collectors and supervisors were trained for three days about the study, communication, respecting the cultural norms of women, detailed review procedures, the informed consent process, and administration of the study questionnaire.

Before collecting the actual data, 5% of the total sample size was pretested, and necessary corrections to the questionnaires were made accordingly. The questionnaire was translated and back-translated from English to Amharic and back to English to check the consistency. At the end of each data collection day, data were checked for completeness and consistency and discussed with the research assistants.

## Data collection instrument and measurement

A structured questionnaire was developed from similar uterine prolapse prevalence studies. The interview question was composed of three main sections. The first two were phase 1, and the third was considered as phase two:

1. socio-demographic variables and obstetric and gynecologic history (14 questions);

2. questions regarding symptoms of uterine prolapsed (6 questions);

3. confirming by vaginal examination whether the women who reported symptomatic prolapse had anatomical prolapse or not and prolapse staging of the present.

Symptomatic POP was assessed by two questions previously used in other studies: do you have a (1) feeling of bulging/pressure or something that seems to be coming down through the vagina or (2) visible mass protruding via the vagina? A woman who had experienced one or both of these problems in the past year was considered as having symptoms of UP, and further questions assessed the duration and associated symptoms. An indication of prolapse based on the questionnaire was referred to as symptomatic prolapse. Women who reported the symptoms of uterine prolapse were referred to Tercha Zonal hospital for pelvic examination to further identify the anatomical prolapse and its stage. A gynaecologist performed pelvic examination using the Pelvic Organ Prolapse Quantification (POP-Q) system at the hospital along with care and treatment.

## Data analysis

Data entry and analysis were conducted using Epi Data v3.2.1 and SPSS v25. Data were cleaned before analysis. The means, frequencies, and percentages were calculated, and bivariate and multivariable logistic regression was carried out to examine the relationships between the independent and dependent variables. Variables with P-values less than 0.25 in bivariate logistic regression were further examined using multivariate logistic regression to investigate

**Table 2. Socio-demographic characteristics of study participants in the Loma Woreda, Dawuro Zone, Ethiopia 2020 (n = 422).**

| Variable | Category | Frequency | Percentage |
|---|---|---|---|
| Marital status | Married | 366 | 86.72 |
| | Divorced | 30 | 7.1 |
| | Widowed | 26 | 6.16 |
| Residence | Urban | 93 | 22.0 |
| | Rural | 329 | 78.0 |
| Educational status | Unable to read and write | 91 | 21.6 |
| | Primary school | 144 | 34.1 |
| | Secondary school | 128 | 30.3 |
| | Higher education | 59 | 14.0 |
| Occupational status | House wife | 313 | 74.2 |
| | Government worker | 65 | 15.4 |
| | Merchant | 36 | 8.5 |
| | Others | 8 | 1.9 |
| Religion | Orthodox | 93 | 22.0 |
| | Protestant | 322 | 76.3 |
| | Others* | 7 | 1.7 |

*Others, musilms, catholics.

associations between the dependent variable and independent variables. Adjusted odds ratio (AOR) was used and a P-value <0.05 was considered statistically significant.

# Results

## Socio-demographic characteristics of study participants

Four hundred and twenty-two women participated in the study, a response rate of 100%. The mean age (+SD) of respondents was 35.4 (±7.99) years, and the mean ages (+SD) at first marriage and first childbirth were 18.14 (±2.151) and 19.94 (±2.921), respectively. The mean numbers of pregnancies and childbirths were 3.94 and 3.80, respectively. Three hundred and sixty-six women (86.5%) were married; over three-quarters (78%) of participants were rural residents. Over half of the participants were unable to read and write or had only primary education (235; 55.7%). Around three-quarters of respondents were housewives (313, 74.2%). More than three fourth of respondents (322, 76.3%) were protestant religion followers (**Table 2**).

## Obstetric and gynecological variables

Half of the participants (217, 51.4%) had a history of home delivery only, while 122 (28.9%) had a history of delivery in a healthcare institution. The largest proportion of study subjects, 89.3% (377), had a history of normal vaginal delivery, while 3.6% and 2.8% had had caesarian sections and operative deliveries, respectively. Only a third of study participants (141) had a history of delivery assisted by health personnel. One hundred and fifty-three (36.3%) and one in ten (10%) women's deliveries were assisted by family/relatives and traditional birth attendants. One in ten of (10.2%) participants had a history of abortion (**Table 3**).

## Prevalence of uterine prolapse

Initial screening with midwives, 28 (6.6%) subjects suspected UP and were referred to doctor examination, and only 25(5.9%) participants confirmed that they had UP. All of these 28 women

**Table 3. Obstetric and gynecological characteristics of study participants in Loma Woreda, Dawuro Zone, Ethiopia 2020 (n = 422).**

| Variable | Category | Frequency | Percentage |
|---|---|---|---|
| Age at first marriage | Less than 18 years | 252 | 59.7 |
| | 18 and above years | 170 | 40.3 |
| Age at first childbirth | Less than 18 years | 113 | 26.8 |
| | 18 and above years | 309 | 73.2 |
| Number of pregnancies | Grand multipara | 158 | 37.4 |
| | Primipara | 59 | 14.0 |
| | Multipara | 205 | 48.6 |
| Place of delivery | Home delivery | 217 | 51.4 |
| | Health institution | 122 | 28.9 |
| | Both home delivery and health institution | 83 | 19.7 |
| Mode of delivery | Normal vaginal delivery | 377 | 89.3 |
| | Operative delivery | 12 | 2.8 |
| | Caesarian section | 15 | 3.6 |
| | Two or more of the above | 18 | 4.3 |
| Birth attendant | Health personnel | 141 | 33.4 |
| | Traditional birth attendant | 42 | 10.0 |
| | Family or relatives | 153 | 36.3 |
| | Two or more of the above | 86 | 20.4 |
| History of abortion | No | 379 | 89.8 |
| | Yes | 43 | 10.2 |

reported a feeling of bulging, pressure, or something coming down from the vagina and nine had a visible mass protruding from the vagina (2.1% of the total study subjects). Of the 28 women who reported symptomatic uterine prolapse, about two-third of (67.86%) had either a feeling of bulging/something coming down in the vagina or a visible mass protruding from the vagina, whereas the other nine (32.14%) reported both bulging and a visible mass protruding from the vagina.

Among 28 women who reported symptomatic prolapse, 25 (5.9% of total) had anatomical prolapse when defined by vaginal checkups and confirmed by doctors. Thus, the prevalence of symptomatic prolapse and anatomical prolapses were 6.6% and 5.9%, respectively. Twenty-five women, 89.29% of those who reported symptoms of UP, had anatomical prolapse. The overall rate of UP was 5.9% (Fig 2).

### Factors associated with uterine prolapse

Multivariate logistic regression conducted identified, age at first marriage, history of abortion, birth attendant who assisted the delivery, and place of delivery were independent factors associated with uterine prolapse (p < 0.05).

Respondents who had a history of abortion were 2.94-times more likely to experience UP (AOR 2.94, 95%CI 1.08, 7.97). Women who were married at age 18 and above were 75% less likely to have uterine prolapse than those who were married before 18 years of age (AOR 0.25, 95%CI 0.07, 0.89). Home delivery was also a risk factor for UP, with women who had a history of home delivery 3.33-times more likely to have UP than other modes of delivery(AOR 3.33, 95%CI 1.21, 9.13). Moreover, women whose delivery was attended by a health professional were 79% less likely to have uterine prolapse than all other birth attendants (AOR 0.21; 95%CI 0.06, 0.71) (**Table 4**).

### Discussion

In this study, age at first marriage, a history of abortion, birth attendant who assisted the delivery, and place of delivery were independent factors associated with uterine prolapse.

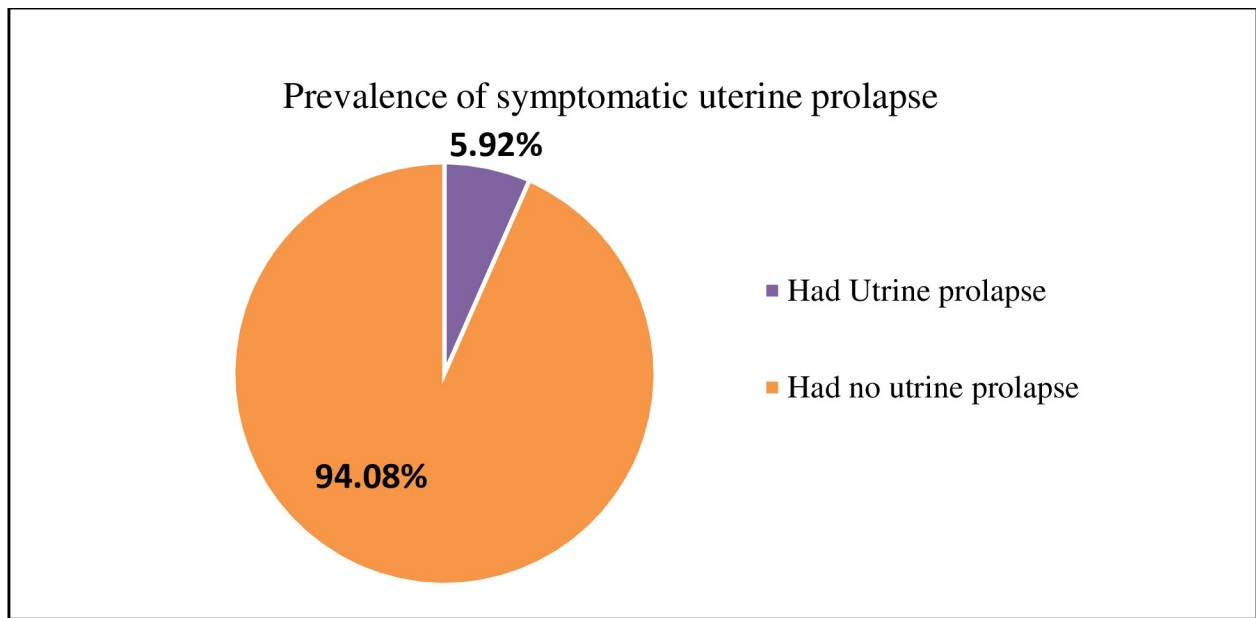

**Fig 2. Prevalence of symptomatic uterine prolapse in Loma Woreda, Dawuro Zone, Ethiopia 2020 (n = 422).**

Here we report the prevalence of symptomatic prolapse in Ethiopia, which was similar to global estimates (3–6%) with similar methodologies [3]. However, our prevalence of 5.9% was less than another study conducted in Nepal, which reported a 13% prevalence [13].

In Iran, the prevalence of UP among women of childbearing age has been reported to be 53.6%, significantly higher than our findings, although both reports early marriage and high parity as the strongest predictors of UP [14]. This significant difference in prevalence might be due to socio-cultural, ethnic, or racial differences. Similarly, the prevalence of UP among married women aged 15–60 in Lebanon was 49.6% [14], which again might be due to socio-

**Table 4. Factors associated with the prevalence of uterine prolapse in the Loma Woreda, Dawuro Zone, Ethiopia 2020 (n = 422).**

| Variable | Category | Uterine prolapse | | COR^ (95%CI) | P-Value | AOR (95%CI) | P-value |
|---|---|---|---|---|---|---|---|
| | | Yes | No | | | | |
| Residence | Urban | 9 | 84 | 0.47 (0.21,1.12) | 0.08 | 0.51 (0.21,1.28) | 0.152 |
| | Rural | 16 | 313 | 1 | | 1 | |
| Educational status | Able to read and write | 3 | 88 | 1 | | 1 | |
| | Unable to read and write | 22 | 309 | 2.08 (0.61,7.14) | 0.24 | 1.15 (0.28,4.66) | 0.836 |
| Age at first mirage | Less than 18 years | 22 | 230 | 1 | | 1 | |
| | 18 and above years | 3 | 167 | 0.199 (0.05,0.64) | 0.007 | 0.25 (0.07,0.89) | 0.033* |
| Place of delivery | Home delivery | 9 | 208 | 1.95 (0.84, 4.53) | 0.117 | 3.33 (1.21,9.13) | 0.020* |
| | Health institution | 4 | 118 | 1 | | 1 | |
| Birth attendant | Health personnel | 4 | 137 | 0.36 (0.12, 1.07) | 0.067 | 0.21 (0.06,0.71) | 0.011* |
| | Others | 21 | 260 | 1 | | 1 | |
| History of abortion | No | 18 | 361 | 1 | | 1 | |
| | Yes | 7 | 36 | 3.9 (1.52, 9.96) | 0.004 | 2.94 (1.08,7.97) | 0.034* |

*Significant at p-value<0.05

^ Crude odd ratio.

cultural, ethnic, or methodological differences. A study from Gambia, West Africa, reported a prevalence rate of 46%, again with parity the strongest risk factor [4]. Besides, a study from Tanzania reported a 64.6% prevalence of anatomical prolapse, substantially higher than the 5.9% anatomical prolapse reported here [8]. However, both studies share home delivery as a risk factor.

Our findings are similar to those from Dabat, Northern Ethiopia, reporting a prevalence of symptomatic prolapse of 6.3% [13], although this study examined the prevalence in all women aged 18 and older rather than reproductive age women alone. Thus, our detected prevalence of UP seems high, given that the prevalence of UP increases with age [5, 13]. Moreover, our detected prevalence of UP is considerably higher than that reported in a study from North and East Ethiopia of only 1% [5]. The difference may be due to socio-cultural variations as well as methodological differences. In Keresa, Eastern Ethiopia, the prevalence of UP in ever-married women was 9.5% [15], a bit higher than reported in the current study.

The findings of this study and a similar study conducted in Wolaita Sodo of south Ethiopia showed age at first marriage and place of delivery were significantly associated with UP [16].

Intensive intermediate Obstetric Critical Care is needed to address maternal complications and their sustainability low resource setting. Studies conducted in Sierra Leone showed an additional cost per QALY of only 10.0; the implementation and one-year running of the case studied obstetric a highly cost-effective innovation [17]. Cost-effective, proven Obstetric Critical Care is needed to address maternal complications.

The limitation of the study was that only symptomatic uterine prolapses were included in this study.

## Conclusion and recommendations

UP is common in reproductive age women in Loma Woreda, Dawuro Zone, Ethiopia. Age at first marriage, place of delivery, birth attendant-assisted delivery, and history of abortion were found to be independent predictors of UP.

We recommend that the health system link primary health care to hospital-set for uterine prolapse treatment programs. Health institution delivery should be encouraged by the local government. Early marriage and unwanted pregnancy need to be prevented through appropriate strategies.

## Supporting information

**S1 Data.**
(SAV)

## Acknowledgments

The author's sincere gratitude goes to the data collectors, supervisors and study participants. The authors also thank Nextgenediting for editorial assistance as part of their Global Initiative.

## Author Contributions

**Conceptualization:** Mengistu Auro Lelu.

**Data curation:** Mengistu Auro Lelu.

**Formal analysis:** Abebe Sorsa Badacho, Mengistu Auro Lelu.

**Funding acquisition:** Mengistu Auro Lelu.

**Investigation:** Abebe Sorsa Badacho, Mengistu Auro Lelu, Zegeye Gelan.

**Methodology:** Abebe Sorsa Badacho, Mengistu Auro Lelu, Deginesh Dawit Woltamo.

**Project administration:** Mengistu Auro Lelu.

**Resources:** Mengistu Auro Lelu.

**Software:** Abebe Sorsa Badacho, Mengistu Auro Lelu.

**Supervision:** Abebe Sorsa Badacho, Mengistu Auro Lelu, Zegeye Gelan.

**Visualization:** Mengistu Auro Lelu.

**Writing – original draft:** Abebe Sorsa Badacho, Mengistu Auro Lelu, Deginesh Dawit Woltamo.

**Writing – review & editing:** Mengistu Auro Lelu, Zegeye Gelan, Deginesh Dawit Woltamo.

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
