## [Decision Letter · Decision Letter 0]

1 Feb 2021

PONE-D-20-40659

Uterine prolapse and associated factors among reproductive age women in Dawro Zone, southwest Ethiopia: a community based cross-sectional study

PLOS ONE

Dear Dr. Sorsa,

Thank you for submitting your manuscript to PLOS ONE. After careful consideration, we feel that it has merit but does not fully meet PLOS ONE’s publication criteria as it currently stands. Therefore, we invite you to submit a revised version of the manuscript that addresses the points raised during the review process.

We look forward to receiving your revised manuscript.

Kind regards,

Claudia Marotta

Academic Editor

PLOS ONE

Journal Requirements:

3. Please amend your manuscript to adhere to our submission guidelines with respect to. Potentially, outmoded terms should be changed to more current, acceptable terminology. Specifically, we recommend that “Vaginal examination” should be changed to more appropriate term(s).

4. You indicated that you had ethical approval for your study. In your Methods section, please ensure you have also stated whether you obtained consent from parents or guardians of the minors included in the study or whether the research ethics committee or IRB specifically waived the need for their consent.

6. Please amend your list of authors on the manuscript to ensure that each author is linked to an affiliation. Authors’ affiliations should reflect the institution where the work was done (if authors moved subsequently, you can also list the new affiliation stating “current affiliation:….” as necessary).

7. Your ethics statement should only appear in the Methods section of your manuscript. If your ethics statement is written in any section besides the Methods, please move it to the Methods section and delete it from any other section. Please ensure that your ethics statement is included in your manuscript, as the ethics statement entered into the online submission form will not be published alongside your manuscript.

8. We note you have included a table to which you do not refer in the text of your manuscript. Please ensure that you refer to Table 1 and 2 in your text; if accepted, production will need this reference to link the reader to the Table.

9.We noticed you have some minor occurrence of overlapping text with the following previous publication(s), which needs to be addressed:

-https://doi.org/10.1007/s00192-007-0375-z

In your revision ensure you cite all your sources (including your own works), and quote or rephrase any duplicated text outside the methods section. Further consideration is dependent on these concerns being addressed.

Additional Editor Comments:

dear Authors follow reviwers suggestion to improve your paper

Reviewers' comments:

Reviewer's Responses to Questions

**Comments to the Author**

1. Is the manuscript technically sound, and do the data support the conclusions?

Reviewer #1: Yes

Reviewer #2: Partly

Reviewer #3: No

Reviewer #4: Yes

2. Has the statistical analysis been performed appropriately and rigorously? 

Reviewer #1: Yes

Reviewer #2: I Don't Know

Reviewer #3: No

Reviewer #4: Yes

3. Have the authors made all data underlying the findings in their manuscript fully available?

Reviewer #1: Yes

Reviewer #2: Yes

Reviewer #3: Yes

Reviewer #4: Yes

4. Is the manuscript presented in an intelligible fashion and written in standard English?

Reviewer #1: Yes

Reviewer #2: Yes

Reviewer #3: No

Reviewer #4: Yes

5. Review Comments to the Author

Reviewer #1: I read with great interest the paper on important topic and from relevant setting

The paper is high quality: both statistical analysis and time of study is excellent

some suggestion

1. Introduction:UP is the most common gynecological health problem contributing to maternal morbidity and

mortality in women of reproductive age in developing countries. It is strongly related with infection pre and post surgery. Infections are the underlying causes in 11% of maternal, and one-fourth of newborn deaths, but the true burden of maternal infections and related complications remains unknown. Add and cite (Maternal caesarean section infection (MACSI) in Sierra Leone: a case-control study. Epidemiol Infect. 2020 Feb 27;148:e40. doi: 10.1017/S0950268820000370. PMID: 32102721; PMCID: PMC7058652.and Epidemiology, Outcomes, and Risk Factors for Mortality in Critically Ill Women Admitted to an Obstetric High-Dependency Unit in Sierra Leone. Am J Trop Med Hyg. 2020 Nov;103(5):2142-2148. doi: 10.4269/ajtmh.20-0623. PMID: 32840199; PMCID: PMC7646769.)

2. Methods and results: no comment are clear

3. Discussion: add and discuss the need for intensive care in Africa especially to address maternal complications and their sustainability as highlighted in previous studies (see Cost-Utility of Intermediate Obstetric Critical Care in a Resource-Limited Setting: A Value-Based Analysis. Ann Glob Health. 2020 Jul 20;86(1):82. doi: 10.5334/aogh.2907. PMID: 32742940; PMCID: PMC7380057.)

Reviewer #2: The manuscript entitled "Uterine prolapse and associated factors among reproductive age women in Dawro Zone, southwest Ethiopia: a community based cross-sectional study" is interesting, but I have some concerns:

1 - The title of the manuscript must be more objective and other details must be mentioned in the methodology. We think the title "Uterine prolapse and associated factors among reproductive age women in southwest Ethiopia" would be better.

2 - We suggest greater clarity of the Abstract, as it represents a mirror of the manuscript.

3 - In Introduction, the authors write a long introduction with two pages and an unnecessary mention of the geographical situation and population of Ethiopia, it would be important to mention only the importance of the study in Ethiopia as it is an underdeveloped or developing country where in some regions there are lack of obstetric care.

4 - This study lost international interest, as it was written with a view to more regional interest. We suggest that the authors review this situation.

5 - Also in Introduction or Discussion, the authors, despite studying uterine prolapse in reproductive age in an underdeveloped country, should stress that total uterine prolapse is more common in post-reproductive age, however carcinoma of the cervix and a prolapsed uterus are common diseases in underdeveloped countries , but their association is quite rare (da Silva BB. Carcinoma of the Cervix in Association with uterine prolapse. Gynecol Oncol

. 2002; 84 (2): 349-50. doi: 10.1006 / gyno.2001.6503).

6 - In Methods, why do the study on genital prolapse only in a community in Ethiopia with a smaller sample, instead of doing a study with a larger sample size and with greater international interest?

7 - In Results, we suggest greater objectivity, long and confusing results, occupying almost five pages.

8 - In discussion, we suggest that the authors discuss the findings of the current study, point by point, with findings in the international literature.

9 - We suggest improving the English edition.

Reviewer #3: About the article “Uterine prolapse and associated factors among reproductive age women in Dawro Zone, southwest Ethiopia: a community based cross-sectional study”:

First, I would like to congratulate the authors by this interesting paper. The authors’ aim was to evaluate the prevalence and factors associated with uterine prolapse among women of reproductive age throw a cross-sectional study. The end point was to analyze risk factors associated with uterine prolapse.

I have some points regard their work:

1- The article needs a native English speaker review;

2- The authors should describe the full words in first passage abbreviations (i.e. AOR, COR);

3- Introduction: The introduction is too long. The authors should shrink it. I suggest that the treatment section can be suppressed;

4- Introduction: last paragraph: I am not sure that they present the prevalence with this cross-sectional study. I understood that they analyzed a few sample of their population. How they make to this sample be representative? They should revise it;

5- Material and methods: “The sample size was calculated by using a single population formula with p=0.5 margin of error and (d) = 0.05 ”. I couldn’t understand how the authors calculated their sample size. What is their populational average used? And the standard deviation? Please, could you explain how it was estimated?;

6- Material and methods: How were the randomization performed?;

7- Material and methods: The authors describe that the patients were selected by an interview done by midwives? If they had symptoms, they were referred to the gynecologist to pelvic examination. The authors should describe as a weakness of the study that only symptomatic uterine prolapses were included in this study;

8- Material and methods: What questionnaire was used? What questions were used in each 3 main sections?;

9- Material and methods: Was the questionnaire validated by a previous study? Please indicate in the text the reference;

10- Material and methods (abstract): I would like to understand why authors included in the multivariate logistic regression the variables from the bivariate logistic regression with P < 0.25 instead of p<0.20. The authors should include this description in the Material and methods area of the main manuscript also;

11- Results: I would like to understand the results presented: “This study identified that the prevalence of symptomatic and anatomical uterine prolapse was 6.6% (28) and 5.9% (25) respectively”. How can be more symptomatic patients that actually anatomical identified uterine prolapse? The authors should explain;

12- Discussion: I suggest that the first paragraph, the authors describe their main objective findings;

13- Abstract Conclusion: The authors may enhance their conclusion. What is the message? What they suggest to their public health as a possible action?;

14- Manuscript Conclusion: The authors should enhance the conclusion to better comprehension.

Cordially

Reviewer #4: 25/01/2021

To

Dr. Claudia Marotta

Managing Editor

PLoS One

Dear Dr. Marotta,

Enclosed please find a review of the manuscript entitled “Uterine prolapse and associated factors among reproductive age women in Dawro Zone, southwest Ethiopia: a community based cross-sectional study” which I am recommending for publication in PLos One after a minor revision.

I have prepared a summary of the study and a list of issues that the authors may want to address.

Best regards,

Salvatore Andrea Mastrolia, MD

Department of Obstetrics and Gynecology

Ospedale dei Bambini "Vittore Buzzi"

Via Lodovico Castelvetro, 32

20154, Milano

Italy

Email mastroliasa@gmail.com

International Fellowship in Advanced Obstetric Training 2013-2016

Soroka University Medical Center

Beer Sheva, ISrael

 

Manuscript #: PONE-D-20-40659

Title: "Uterine prolapse and associated factors among reproductive age women in Dawro Zone, southwest Ethiopia: a community based cross-sectional study”

Comments for the author

General comments: I am happy of the chance to review this manuscript focusing on a subject that is still not adequately assessed, not only in developing countries but also in high income countries. I read the manuscript with interest and here is a list of minor concerns that the authors may want to address:

1) I would suggest a linguistic revision in order to improve the readability of the manuscript

2) In the Methods section the Authors report that variables with P-values less than 0.25 in bivariate logistic regression were further examined using multivariate logistic regression to investigate associations between the dependent variable and independent variables. How was this value of 0.25 chosen?

3) The Discussion section can be slightly extended assessing potential physiopathologic explanations for the associations found by the Authors.

6. PLOS authors have the option to publish the peer review history of their article (what does this mean?). If published, this will include your full peer review and any attached files.

Reviewer #1: No

Reviewer #2: No

Reviewer #3: **Yes: **Ricardo Pedrini Cruz

Reviewer #4: **Yes: **Salvatore Andrea Mastrolia

---

## [Author Response · Author response to Decision Letter 0]

24 May 2021

Authors response: Dear Editor : Thank you so much, The manuscript meets PLOS ONE stylerequirements. 

 Authors response: Dear Editor : Thank you so much, questionnaire developed included both original language and English, as Supporting Information.

3. Please amend your manuscript to adhere to our submission guidelines with respect to. Potentially, outmoded terms should be changed to more current, acceptable terminology. Specifically, we recommend that “Vaginal examination” should be changed to more appropriate term(s).

Authors response: Dear Editor : Thank you so much, “Vaginal examination” changed to vaginal checkup. 

4. You indicated that you had ethical approval for your study. In your Methods section, please ensure you have also stated whether you obtained consent from parents or guardians of the minors included in the study or whether the research ethics committee or IRB specifically waived the need for their consent.

Authors response: Dear Editor : Thank you so much, No minors included in the study. 

Authors response: Dear Editor : Thank you so much, I have an ORCID iD 0000-0002-1377-8298

6. Please amend your list of authors on the manuscript to ensure that each author is linked to an affiliation. Authors’ affiliations should reflect the institution where the work was done (if authors moved subsequently, you can also list the new affiliation stating “current affiliation:….” as necessary).

Authors response: Dear Editor : Thank you so much, list of authors amended on the manuscript.

7. Your ethics statement should only appear in the Methods section of your manuscript. If your ethics statement is written in any section besides the Methods, please move it to the Methods section and delete it from any other section. Please ensure that your ethics statement is included in your manuscript, as the ethics statement entered into the online submission form will not be published alongside your manuscript.

 Authors response: Dear Editor : Thank you so much, ethics statement included in the Methods section and deleted from other section.

8. We note you have included a table to which you do not refer in the text of your manuscript. Please ensure that you refer to Table 1 and 2 in your text; if accepted, production will need this reference to link the reader to the Table.

Authors response: Dear Editor : Thank you so much, we ensure that Table 1 and 2 refer in the text

9.We noticed you have some minor occurrence of overlapping text with the following previous publication(s), which needs to be addressed:

-https://doi.org/10.1007/s00192-007-0375-z

In your revision ensure you cite all your sources (including your own works), and quote or rephrase any duplicated text outside the methods section. Further consideration is dependent on these concerns being addressed.

Authors response: Dear Editor : Thank you so much, we revised and cited all sources. 

 

Reviewer #1

I read the paper critically on an important topic and from a relevant setting. The paper is high quality: both statistical analysis and time of study is excellent some suggestion.

Authors' response #2: Dear Reviewer, Thank you so much; reading critically and reviewing it and your suggestion. 

1. Introduction: UP is the most common gynaecological health problem contributing to maternal morbidity and mortality in women of reproductive age in developing countries. It is strongly related to infection pre and post-surgery. Infections are the underlying causes in 11% of maternal and one-fourth of newborn deaths, but the true burden of maternal infections and related complications remains unknown. Add and cite (Maternal caesarean section infection (MACSI) in Sierra Leone: a case-control study. Epidemiol Infect. 2020 Feb 27;148:e40. doi: 10.1017/S0950268820000370. PMID: 32102721; PMCID: PMC7058652.and Epidemiology, Outcomes, and Risk Factors for Mortality in Critically Ill Women Admitted to an Obstetric High-Dependency Unit in Sierra Leone. Am J Trop Med Hyg. 2020 Nov;103(5):2142-2148. doi: 10.4269/ajtmh.20-0623. PMID: 32840199; PMCID: PMC7646769.)

Authors' response: Dear Reviewer, Thank you so much; We included. "UP is the most common gynaecological health problem contributing to maternal morbidity and mortality in women of reproductive age in developing countries. It is strongly related to infection pre and post-surgery. Infections are the underlying causes in 11% of maternal and one-fourth of newborn deaths, but the actual burden of maternal infections and related complications remains unknown. (Di Gennaro et al., 2020, Marotta et al., 2020)."

 Please see lines 69 to 73.

Comment #2. Methods and results: no comment are clear

Authors' response #2: Dear Reviewer, Thank you so much;

Comment #3 Discussion: add and discuss the need for intensive care in Africa, especially to address maternal complications and sustainability, as highlighted in previous studies (see Cost-Utility of Intermediate Obstetric Critical Care in a Resource-Limited Setting: A Value-Based Analysis. Ann Glob Health. 2020 Jul 20;86(1):82. doi: 10.5334/aogh.2907. PMID: 32742940; PMCID: PMC7380057.)

Authors' response to Comment # 3: Dear Reviewer, Thank you so much; We included "Intensive intermediate Obstetric Critical Care is needed to address maternal complications and their sustainability low resource setting. Studies conducted in Sierra Leone showed an additional cost per QALY of only €10.0; the implementation and one-year running of the case studied obstetric a highly cost-effective innovation(Marotta et al., 2020)." Please see Linens 229 to 233. 

Reviewer #2: The manuscript entitled "Uterine prolapse and associated factors among reproductive-age women in Dawro Zone, southwest Ethiopia: a community based cross-sectional study" is interesting, but I have some concerns:

Authors' response #2: Dear Reviewer, Thank you so much; for your concerns 

Comment #1 - The manuscript title must be more objective, and other details must be mentioned in the methodology. We think the title "Uterine prolapse and associated factors among reproductive age women in southwest Ethiopia" would be better.

Authors' response: Dear Reviewer, Thank you so much; we modified the manuscript title "Uterine prolapse and associated factors among reproductive-age women in southwest Ethiopia" Please see line 2 1ine 3. 

Comment # 2 - We suggest greater clarity of the Abstract, as it represents a mirror of the manuscript.

Authors' response # 2: Dear Reviewer, Thank you so much; we clarity in the abstract of manuscript. 

Comment # 3 - In the Introduction, the authors write a long introduction with two pages and an unnecessary mention of the geographical situation and population of Ethiopia, it would be important to mention only the importance of the study in Ethiopia as it is an underdeveloped or developing country where in some regions there are lack of obstetric care.

Authors' response# 3: Dear Reviewer, Thank you so much; we removed unnecessary mention of geographic situation."According to a projection from the 2007 national census, in 2019, Ethiopia has nearly 110 million inhabitants. The women in the reproductive age group constitute 23.4%. Thus, nearly 25.5 million Ethiopian women are between 15 and 49 years of age. These women, in one or other ways, are affected by the burden of uterine prolapse". 

Comment # 4 - This study lost international interest, as it was written with a view to more regional interest. We suggest that the authors review this situation.

Authors' response# 4: Dear Reviewer, Thank you so much; we reviewed the situation. 

Comment # 5 - Also in Introduction or Discussion, the authors, despite studying uterine prolapse in reproductive age in an underdeveloped country, should stress that total uterine prolapse is more common in post-reproductive age, however carcinoma of the cervix and a prolapsed uterus are common diseases in underdeveloped countries , but their association is quite rare (da Silva BB. Carcinoma of the Cervix in Association with uterine prolapse. Gynecol Oncol

. 2002; 84 (2): 349-50. doi: 10.1006 / gyno.2001.6503).

Authors' response# 5: Dear Reviewer, Thank you so much; we included in discussion part “Uterine prolapse is more common in post-reproductive age, however carcinoma of the cervix and a prolapsed uterus are common diseases in underdeveloped countries , but their association is quite rare (da Silva BB. Carcinoma of the Cervix in Association with uterine prolapse. Gynecol Oncol. 2002; 84 (2): 349-50. doi: 10.1006 / gyno.2001.6503). 

Please see line 84 to 86. 

Comment # 6 - In Methods, why do the study on genital prolapse only in a community in Ethiopia with a smaller sample instead of studying with a larger sample size and greater international interest?

Authors' response#6: Dear Reviewer, Thank you so much; we accept your comment; doing a study with a larger sample size and with greater international interest is more appropriate but due to financial constraints, we focused our study on genital prolapse. 

We are planning to conduct an extensive study in future. 

Comment # 7 - In Results, we suggest greater objectivity, long and confusing results, occupying almost five pages.

Authors' response#7: Dear reviewer; Thank you so much, we made results most concince and clean by removing all unnecessary statements with in three pages. Please see lines 160 to 203

Comment # 8 - In discussion, we suggest that the authors discuss the findings of the current study, point by point, with findings in the international literature.

Authors' response#8 : Dear reviewer; Thank you for your comment. We traid to to discuss point by point with finding from international literature. 

Comment # 9 - We suggest improving the English edition.

Authors' response #9: Dear reviewer; Thank you so much. We made a language edition by international language editors—Nextgenediting for editorial assistance as part of their Global Initiative.

 

Reviewer #3: 

About the article "Uterine prolapse and associated factors among reproductive age women in Dawro Zone, southwest Ethiopia: a community based cross-sectional study":

First, I would like to congratulate the authors by this interesting paper. The authors' aim was to evaluate the prevalence and factors associated with uterine prolapse among women of reproductive age throw a cross-sectional study. The end point was to analyze risk factors associated with uterine prolapse.

Authors' response: Thank you so much

I have some points regard their work:

1- The article needs a native English speaker review;

Authors' response#1: Dear reviewer, Thank you so much. We made language edition by international language editors. Nextgenediting for editorial assistance as part of their Global Initiative.

2- The authors should describe the full words in first passage abbreviations (i.e. AOR, COR);

Authors' response#2: Thank you so much. We fully desrcibed the full words in first passage abbreviation. Please see line 49.

3- Introduction: The introduction is too long. The authors should shrink it. I suggest that the treatment section can be suppressed; repressed 

Authors' response#3: Thank you; we suppressed the treatment section. The treatment of UP depends on the severity of the prolapse, the woman's general health, age and desire to have children. First and second-degree prolapse is usually treated using ring pessaries inserted into the vagina to stretch the vaginal walls, often used in combination with pelvic floor exercises. The prolapse above the second degree is often treated by surgery (Elsayed, 2016).

4- Introduction: last paragraph: I am not sure that they present the prevalence with this cross-sectional study. I understood that they analyzed a few sample of their population. How they make to this sample be representative? They should revise it;

Authors' response#4: Thank you so much; we determined sample size based on scientific method size determination which gave us enough sample size. Participants were randomly selected in the study. 

5- Material and methods: "The sample size was calculated by using a single population formula with p=0.5 margin of error and (d) = 0.05". I couldn't understand how the authors calculated their sample size. What is their populational average used? And the standard deviation? Please, could you explain how it was estimated?;

Authors' response#5: Thank you so much. We calculated sample size by using a confidence interval approach for a single population proportion with 50% since no published study regarding prevalence of UP in the country settled up. Therefore, we used to assume a proportion of 50% to get the maximum sample to be representative.

6- Material and methods: How were the randomization performed?;

Authors' response#6: Thank you so much. From the total 28 kebeles in the Loma district, 30% of kebeles, i.e., a total of eight kebeles, were selected by lottery method. The assumption was to divide the total estimated sample size to the households of each kebele according to the proportion they contribute to the total study subjects. 

We allocated sample proportion for the selected eight kebeles based on sample size. Out of an estimated 422 participants, sample size was adjusted proportionally for the households of the selected eight kebeles. Then, households were selected by systematic random sampling method, i.e., selecting households at a fixed interval throughout a household list from a registration book of health post as provided by Health Extension Workers(HEWs) working at Health Post in each respective kebele. 

7- Material and methods: The authors describe that an interview done by midwives selected the patients? If they had symptoms, they were referred to the gynaecologist for a pelvic examination. The authors should describe as a weakness of the study that only symptomatic uterine prolapses were included in this study;

Authors' response#7: Thank you so much. We described the li mitation of the study in the limitation of the study section. Please see lines 234 to 235.

8- Material and methods: What questionnaire was used? What questions were used in each 3 main sections?; 

Authors' response#8: The interview question was composed of three main sections; the first two were phase 1 and the third was considered as phase two:

1) Socio-demographic variables and Obstetric and gynaecologic history (14 questions), 

2) Questions regarding symptoms of uterine prolapsed (6questions), 

3) The third section included two items; confirming by vaginal examination whether the women who reported symptomatic prolapse had anatomical prolapse or not, and the staging the prolapse was done then. 

9- Material and methods: Was the questionnaire validated by a previous study? Please indicate in the text the reference;

Authors' response#9 : Thanks so much; we are not sure that the questionnaire was not validated in the previous study.

10- Material and methods (abstract): I would like to understand why the authors included in the multivariate logistic regression the variables from the bivariate logistic regression with P < 0.25 instead of p<0.20. The authors should include this description in the Material and methods area of the main manuscript also;

Authors' response #10: Thank you so much; We included variables with p< 0.25 for candidate variable for the final model to increase the chance of the variable became a significant predictor variable. We included the description in the material and methods section. 

11- Results: I would like to understand the results presented: "This study identified that the prevalence of symptomatic and anatomical uterine prolapse was 6.6% (28) and 5.9% (25) respectively". How can be more symptomatic patients that actually anatomical identified uterine prolapse? The authors should explain;

Authors' response #11: Dear reviewer, thank you so much, the data collectors were trained midwifes and creafully assessed symptomatic patients and most of the symptomatic patients identified as anatomically identified utrine prolapse by doctors. 

12- Discussion: I suggest that the first paragraph, the authors describe their main objective findings;

Authors' response #12: Dear reviewer, thank you so much, we included the statement “In this study; age at first marriage, a history of abortion, birth attendant who assisted the delivery, and place of delivery were independent factors associated with uterine prolapse.” 

13- Abstract Conclusion: The authors may enhance their conclusion. What is the message? What they suggest to their public health as a possible action?;

Authors' response #13: Dear reviewer, thank you so much. We enhance the conclusion, including the message: We recommend that the health system link primary health care to hospital-set for uterine prolapse treatment programs. Health institution delivery should be encouraged by the local government. Early marriage and unwanted pregnancy need to be prevented through appropriate strategies. Please See line 49 to 52. 

14- Manuscript Conclusion: The authors should enhance the conclusion to better comprehension.

Cordially

Authors' response #14: Dear reviewer, thank you so much. We enhance the conclusion to better comprehension, including the "We recommend that the health system is linking primary health care to hospital-set for uterine prolapse treatment programs. Health institution delivery should be encouraged by the local government. Early marriage and unwanted pregnancy need to be prevented through appropriate strategies." Please see lines 240- 243.

 

Reviewer #4: 25/01/2021

Enclosed, please find a review of the manuscript entitled "Uterine prolapse and associated factors among reproductive-age women in Dawro Zone, southwest Ethiopia: a community based cross-sectional study" which I am recommending for publication in PLoS One after a minor revision. I have prepared a summary of the study and a list of issues that the authors may want to address

Title: "Uterine prolapse and associated factors among reproductive-age women in Dawro Zone, southwest Ethiopia: a community based cross-sectional study."

Comments for the author

General comments: I am happy of the chance to review this manuscript focusing on a subject that is still not adequately assessed, not only in developing countries but also in high income countries. I read the manuscript with interest and here is a list of minor concerns that the authors may want to address:

Authors' response #: Dear Reviewer, thank so much 

1) I would suggest a linguistic revision in order to improve the readability of the manuscript

Authors' response#1: Dear Reviewer, Thank you so much. We made a language edition by international language editors—Nextgenediting for editorial assistance as part of their Global Initiative.

2) In the Methods section, the Authors report that variables with P-values less than 0.25 in bivariate logistic regression were further examined using multivariate logistic regression to investigate associations between the dependent variable and independent variables. How was this value of 0.25 chosen?

Authors' response #2: Dear Reviewer, Thank you so much; We included variables with p< 0.25 for the candidate variable for the final model to increase the chance of the variable became a significant predictor variable. We included the description in the material and methods section. 

3) The Discussion section can be slightly extended to assess potential physiopathologic explanations for the associations found by the Authors.

Authors' response #2: Dear Reviewer, Thank you so much; We extended the discussion part.

---

## [Decision Letter · Decision Letter 1]

16 Jun 2021

PONE-D-20-40659R1

Uterine prolapse and associated factors among reproductive-age women in south-west Ethiopia : a community-based cross-sectional study

PLOS ONE

Dear Dr. badacho,

Thank you for submitting your manuscript to PLOS ONE. After careful consideration, we feel that it has merit but does not fully meet PLOS ONE’s publication criteria as it currently stands. Therefore, we invite you to submit a revised version of the manuscript that addresses the points raised during the review process.

We look forward to receiving your revised manuscript.

Kind regards,

Richard Kao Lee, M.D.

Academic Editor

PLOS ONE

Journal Requirements:

Additional Editor Comments (if provided):

Reviewers' comments:

Reviewer's Responses to Questions

**Comments to the Author**

1. If the authors have adequately addressed your comments raised in a previous round of review and you feel that this manuscript is now acceptable for publication, you may indicate that here to bypass the “Comments to the Author” section, enter your conflict of interest statement in the “Confidential to Editor” section, and submit your "Accept" recommendation.

Reviewer #1: All comments have been addressed

Reviewer #2: All comments have been addressed

Reviewer #3: All comments have been addressed

2. Is the manuscript technically sound, and do the data support the conclusions?

Reviewer #1: Yes

Reviewer #2: (No Response)

Reviewer #3: No

3. Has the statistical analysis been performed appropriately and rigorously? 

Reviewer #1: Yes

Reviewer #2: No

Reviewer #3: No

4. Have the authors made all data underlying the findings in their manuscript fully available?

Reviewer #1: Yes

Reviewer #2: (No Response)

Reviewer #3: No

5. Is the manuscript presented in an intelligible fashion and written in standard English?

Reviewer #1: Yes

Reviewer #2: No

Reviewer #3: Yes

6. Review Comments to the Author

Reviewer #1: Authors wrote an important paper on important issue. Maternal mortality is a big global health issue

Reviewer #2: The manuscript entitled "Uterine prolapse and associated factors among reproductive age women in Dawro Zone, southwest Ethiopia: a community based cross-sectional study" R1, arouses interest, but I have some regards:

1 - Although the authors show a manuscript on "Uterine prolapse and associated factors among reproductive age women in Dawro Zone, southwest Ethiopia", they fail to stress that, globally, genital prolapse preferentially affects multiparous and elderly women (Hendrix et al. Am J Obstet Gynecol.2002;186(6):1160-6. doi: 10.1067/mob.2002.123819).

2 - Although in Methods it is stated that the staging of uterine prolapse was performed by a gynecologist based on the POP-Q classification, in Background, line 67, the authors mention an old staging of uterine prolapse, when they should have mentioned the updated staging, International Urogynecological Association (IUGA)/International Continence Society (ICS) joint report on the terminology for female pelvic floor dysfunctio. Wein AJ. J Urol . 2011;185(5):1812. doi: 10.1016/S0022-5347(11)60226-7).

3 - In methods, I think the authors exploit a lot of unnecessary geographic information and human resources, such as, location of the city where the study was done and how many kilometers away from the capital of Turkey. Number of beds, Hospital nurses, and so on. There should be a better suitability of the study for publication.

4 - The sample number is small and from this, the prevalence of genital prolapse in only 28 women.

5 - In Results, we look at a lot of numerical information. We suggest greater clarity.

6 - The discussion is short in that the results are not well discussed, with few reference citations. We suggest discussing the results point by point.

Reviewer #3: About the revised article “Uterine prolapse and associated factors among reproductive age women in southwest Ethiopia”: The authors’ aim was to evaluate the prevalence and factors associated with uterine prolapse among women of reproductive age throw a cross-sectional study. The end point was to analyze risk factors associated with uterine prolapse.

First, I would like to congratulate the authors by their effort in reviewing the reviewers’ statements. Again, I congratulate the authors by the study importance. It is increasing in our population the desire of “humanized childbearing”, with more deliveries done at home, without adequate professional help.

Each point with the revision needed (in my humble opinion):

1- The article needs a native English speaker review; - done

2- The authors should describe the full words in first passage abbreviations (i.e. AOR, COR);

“Authors' response#2: Thank you so much. We fully desrcibed the full words in first passage abbreviation. Please see line 49.”

Done

3- Introduction: The introduction is too long. The authors should shrink it. I suggest that the treatment section can be suppressed; - done partially

The introduction remains too long. It would be better that they cut unnecessary parts.

4- Introduction: last paragraph: I am not sure that they present the prevalence with this cross-sectional study. I understood that they analyzed a few sample of their population. How they make to this sample be representative? They should revise it;

“Authors' response#4: Thank you so much; we determined sample size based on scientific method size determination which gave us enough sample size. Participants were randomly selected in the study.”

Thank you, but the answer is still lacking to me, as you can see the next question.

Also, the authors described that the screening was done by trained midwifes, finding 28 subjects that were suspicion of having UP. After doctor examination, only 25 confirmed. Also, the authors didn’t comment, but not even all UP the patients presents with symptoms. So, they can only estimate the incidence of symptomatic UP. The prevalence of UP is still not known, because only a few subjects were examined.

5- Material and methods: “The sample size was calculated by using a single population formula with p=0.5 margin of error and (d) = 0.05 ”. I couldn’t understand how the authors calculated their sample size. What is their populational average used? And the standard deviation? Please, could you explain how it was estimated?;

“Authors' response#5: Thank you so much. We calculated sample size by using a confidence interval approach for a single population proportion with 50% since no published study regarding prevalence of UP in the country settled up. Therefore, we used to assume a proportion of 50% to get the maximum sample to be representative.”

Thank you for the explanation. I am afraid that their explanation wasn’t enough to understand their sample size calculation. By the best of my limit knowledge, to calculate a sample by confidence interval approach for a single population proportion, you must know your standard deviation, not presented in the method area.

Also, if they assume a 50% margin of error, that is a chance of 50% that their null hypothesis is true. I think that the authors tried to use the sample calculation of other publication (Silwal M, Gurung, G., Shrestha, N., Gurung, A. and Ojha, S.: Prevalence and Factors Affecting Women with Uterine Prolapse in Lekhnath, Kaski, Nepal. , 9(2), pp.52-57. Journal of Gandaki Medical College-Nepal 2016, 9(2):6.). I suggest that they revise it with and epidemiologist or a mathematician.

6- Material and methods: How were the randomization performed?;

“Authors' response#6: Thank you so much. From the total 28 kebeles in the Loma district, 30% of kebeles, i.e., a total of eight kebeles, were selected by lottery method. The assumption was to divide the total estimated sample size to the households of each kebele according to the proportion they contribute to the total study subjects. We allocated sample proportion for the selected eight kebeles based on sample size. Out of an estimated 422 participants, sample size was adjusted proportionally for the households of the selected eight kebeles. Then, households were selected by systematic random sampling method, i.e., selecting households at a fixed interval throughout a household list from a registration book of health post as provided by Health Extension Workers(HEWs) working at Health Post in each respective kebele.”

The explanation of randomization must be described in the “Material and methods” area.

7- Material and methods: The authors describe that the patients were selected by an interview done by midwives? If they had symptoms, they were referred to the gynecologist to pelvic examination. The authors should describe as a weakness of the study that only symptomatic uterine prolapses were included in this study;

Authors' response#7: Thank you so much. We described the li mitation of the study in the limitation of the study section. Please see lines 234 to 235.

Done. I suggest that the authors present as a new paragraph only. It does not need that spotlight as a new subdivision in the discussion.

8- Material and methods: What questionnaire was used? What questions were used in each 3 main sections?;

“Authors' response#8: The interview question was composed of three main sections; the first two were phase 1 and the third was considered as phase two:

1) Socio-demographic variables and Obstetric and gynaecologic history (14 questions), 2) Questions regarding symptoms of uterine prolapsed (6questions),

3) The third section included two items; confirming by vaginal examination whether the women who reported symptomatic prolapse had anatomical prolapse or not, and the staging the prolapse was done then.”

Thank you for the intent to answer the question. Unfortunately, the authors copy the explanation that was already in their article. The questions were not revealed in the article, neither in their review. It is not clear to readers what criteria was used to select patients. What questions were used? It is important to describe all the method used in the paper. Maybe their questions can be useful to others in the screening of UP.

9- Material and methods: Was the questionnaire validated by a previous study? Please indicate in the text the reference;

“Authors' response#9 : Thanks so much; we are not sure that the questionnaire was not validated in the previous study.”

The question 9 is related to question 8. They do not show questions used. Also, they were not previous used in other articles. So, how they are so sure that their questions are adequate to screening of their sample?

10- Material and methods (abstract): I would like to understand why authors included in the multivariate logistic regression the variables from the bivariate logistic regression with P < 0.25 instead of p<0.20. The authors should include this description in the Material and methods area of the main manuscript also;

“Authors' response #10: Thank you so much; We included variables with p< 0.25 for candidate variable for the final model to increase the chance of the variable became a significant predictor variable. We included the description in the material and methods section.”

Thank you for the explanation. Unfortunately, their desire to make variables more prone to be significant biased their study. I suggest that it be analyze again, with the correct parameters often used in medical literature.

11- Results: I would like to understand the results presented: “This study identified that the prevalence of symptomatic and anatomical uterine prolapse was 6.6% (28) and 5.9% (25) respectively”. How can be more symptomatic patients that actually anatomical identified uterine prolapse? The authors should explain;

“Authors' response #11: Dear reviewer, thank you so much, the data collectors were trained midwifes and creafully assessed symptomatic patients and most of the symptomatic patients identified as anatomically identified utrine prolapse by doctors.”

Thank you, unfortunately the question remains: how they find more symptoms than they find the pathology? The symptomatology is not a diagnosis. So, in the initial screening with midwifes, there were 28 subjects that were suspicion of having UP. After doctor examination, only 25 confirmed. They should make the corrections pointed.

12- Discussion: I suggest that the first paragraph, the authors describe their main objective findings;

“Authors' response #12: Dear reviewer, thank you so much, we included the statement “In this study; age at first marriage, a history of abortion, birth attendant who assisted the delivery, and place of delivery were independent factors associated with uterine prolapse.”

Perfect done.

13- Abstract Conclusion: The authors may enhance their conclusion. What is the message? What they suggest to their public health as a possible action?;

“Authors' response #13: Dear reviewer, thank you so much. We enhance the conclusion, including the message: We recommend that the health system link primary health care to hospital-set for uterine prolapse treatment programs. Health institution delivery should be encouraged by the local government. Early marriage and unwanted pregnancy need to be prevented through appropriate strategies. Please See line 49 to 52.”

Done.

14- Manuscript Conclusion: The authors should enhance the conclusion to better comprehension.

“Authors' response #14: Dear reviewer, thank you so much. We enhance the conclusion to better comprehension, including the "We recommend that the health system is linking primary health care to hospital-set for uterine prolapse treatment programs. Health institution delivery should be encouraged by the local government. Early marriage and unwanted pregnancy need to be prevented through appropriate strategies." Please see lines 240- 243.”

Done.

15- Introduction: They erased lines 118-119: “According to a projection from the 2007 national census, Ethiopia had nearly 110 million 119 inhabitants in 2019, with 23.4% of women of reproductive age.”. The subsequent phase must be adjusted: “Given this massive potentially…”

16- Discussion, lines 250-251: Symptomatic prolapse in your sample wasn’t 6.6%. If the patients had gynecological examination by doctors and they diagnosticated only 25 of the total sample of 422, your result is 5.9%. There are 3 patients screened to UP that had not the diagnosis of UP made.

17- Figure 2: The same as the previous: the presence of uterine prolapse was only in 25 patients (5.9%). The authors should revise it. A questionary cannot be the gold standard to UP diagnosis. The clinical evaluation is the gold standard.

18- Table 1: marital status: Is missing data of 1 patient. They should revise.

19- Table 1: religion: Is missing data of 1.7% of their sample. They should revise.

Cordially

7. PLOS authors have the option to publish the peer review history of their article (what does this mean?). If published, this will include your full peer review and any attached files.

Reviewer #1: No

Reviewer #2: No

Reviewer #3: **Yes: **Ricardo Pedrini Cruz

---

## [Author Response · Author response to Decision Letter 1]

27 Aug 2021

PONE-D-20-40659R1

Uterine prolapse and associated factors among reproductive-age women in Dawro Zone, southwest Ethiopia: a community based cross-sectional study

PLOS ONE

Subject: Point by point response to reviewers' comments

 Dear PLOS ONE Editorial Manager,

Please kindly find the submitted revised manuscript with ID number PONE-D-20-40659R1, entitled "Uterine prolapse and associated factors among reproductive-age women in Dawro Zone, southwest Ethiopia: a community based cross-sectional study" point to point response to reviewer's comment for kindly consideration. 

With kind regards,

Abebe Sorsa (corresponding Author)

Authors point by point response to Reviewer's Responses to Questions

Comments to the Author

1. If the authors have adequately addressed your comments raised in a previous round of review and you feel that this manuscript is now acceptable for publication, you may indicate that here to bypass the "Comments to the Author" section, enter your conflict of interest statement in the "Confidential to Editor" section, and submit your "Accept" recommendation.

Reviewer #1: All comments have been addressed

Authors' response: Dear Reviewer, Thank you so much

Reviewer #2: All comments have been addressed

Authors' response: Dear Reviewer, Thank you so much

Reviewer #3: All comments have been addressed

Authors' response: Dear Reviewer, Thank you so much

2. Is the manuscript technically sound, and do the data support the conclusions?

Reviewer #1: Yes

Reviewer #2: (No Response)

Reviewer #3: No

3. Has the statistical analysis been performed appropriately and rigorously?

Reviewer #1: Yes

Reviewer #2: No

Reviewer #3: No

4. Have the authors made all data underlying the findings in their manuscript fully available?

Reviewer #1: Yes

Reviewer #2: (No Response)

Reviewer #3: No

5. Is the manuscript presented in an intelligible fashion and written in standard English?

Reviewer #1: Yes

Reviewer #2: No

Reviewer #3: Yes

6. Review Comments to the Author

Reviewer #1: Authors wrote an important paper on important issue. Maternal mortality is a big global health issue

Reviewer #2: The manuscript entitled "Uterine prolapse and associated factors among reproductive age women in Dawro Zone, southwest Ethiopia: a community based cross-sectional study" R1, arouses interest, but I have some regards:

Authors' response: Dear Reviewer, Thank you so much for reviewing it and your suggestions 

1 - Although the authors show a manuscript on "Uterine prolapse and associated factors among reproductive age women in Dawro Zone, southwest Ethiopia", they fail to stress that, globally, genital prolapse preferentially affects multiparous and elderly women (Hendrix et al. Am J Obstet Gynecol.2002;186(6):1160-6. doi: 10.1067/mob.2002.123819).

Authors' response: Dear Reviewer, Thank you so much: Parity and obesity were strongly associated with increased risk for uterine prolapse, cystocele, and rectocele (Hendrix et al. Am J Obstet Gynecol.2002;186(6):1160-6. doi: 10.1067/mob.2002.123819).But the objective of our study focused to assess the prevalence of UP and associated factors among reproductive age group. 

2 - Although in Methods it is stated that the staging of uterine prolapse was performed by a gynecologist based on the POP-Q classification, in Background, line 67, the authors mention an old staging of uterine prolapse, when they should have mentioned the updated staging, International Urogynecological Association (IUGA)/International Continence Society (ICS) joint report on the terminology for female pelvic floor dysfunctio. Wein AJ. J Urol . 2011;185(5):1812. doi: 10.1016/S0022-5347(11)60226-7).

Authors' response: Dear Reviewer, Thank you so much:

3 - In methods, I think the authors exploit a lot of unnecessary geographic information and human resources, such as, location of the city where the study was done and how many kilometers away from the capital of Turkey. Number of beds, Hospital nurses, and so on. There should be a better suitability of the study for publication.

Authors' response: Dear Reviewer, Thank you so much: unnecessary geographic information and human resource location have been removed from the manuscript. Please refer to lines 105 to 113 on track changed manuscript. 

4 - The sample number is small and from this, the prevalence of genital prolapse in only 28 women.

Authors' response: Dear Reviewer, Thank you so much: the sample size was determined using the scientific method of sample size calculation. We agree with your comment that it would be better to use large sample, but it was not feasible for this study. 

5 - In Results, we look at a lot of numerical information. We suggest greater clarity.

Authors' response: Dear Reviewer, Thank you so much: we revised the manuscript for greater clarity 

6 - The discussion is short in that the results are not well discussed, with few reference citations. We suggest discussing the results point by point.

Authors' response: Dear Reviewer, Thank you so much:

Reviewer #3: About the revised article "Uterine prolapse and associated factors among reproductive age women in southwest Ethiopia": The authors' aim was to evaluate the prevalence and factors associated with uterine prolapse among women of reproductive age throw a cross-sectional study. The end point was to analyze risk factors associated with uterine prolapse.

First, I would like to congratulate the authors by their effort in reviewing the reviewers' statements. Again, I congratulate the authors by the study importance. It is increasing in our population the desire of "humanized childbearing", with more deliveries done at home, without adequate professional help.

Each point with the revision needed (in my humble opinion):

1- The article needs a native English speaker review; - done

Authors' response: Dear Reviewer, Thank you so much: for your suggestion 

2- The authors should describe the full words in first passage abbreviations (i.e. AOR, COR);

"Authors' response#2: Thank you so much. We fully desrcibed the full words in first passage abbreviation. Please see line 49."

Done

Authors' response: Dear Reviewer, Thank you so much:

3- Introduction: The introduction is too long. The authors should shrink it. I suggest that the treatment section can be suppressed; - done partially

The introduction remains too long. It would be better that they cut unnecessary parts.

Authors' response: Dear Reviewer, Thank you so much: we revised the introduction part.

4- Introduction: last paragraph: I am not sure that they present the prevalence with this cross-sectional study. I understood that they analyzed a few sample of their population. How they make to this sample be representative? They should revise it;

"Authors' response#4: Thank you so much; we determined sample size based on scientific method size determination which gave us enough sample size. Participants were randomly selected in the study."

Thank you, but the answer is still lacking to me, as you can see the next question.

Also, the authors described that the screening was done by trained midwifes, finding 28 subjects that were suspicion of having UP. After doctor examination, only 25 confirmed. Also, the authors didn't comment, but not even all UP the patients presents with symptoms. So, they can only estimate the incidence of symptomatic UP. The prevalence of UP is still not known, because only a few subjects were examined.

Authors' response: Dear Reviewer, Thank you so much: even though we determined sample size using the scientific method of sample size calculation the subjected examined for UP was few that affected the prevalence of UP among the reproductive age group. 

5- Material and methods: "The sample size was calculated by using a single population formula with p=0.5 margin of error and (d) = 0.05". I couldn't understand how the authors calculated their sample size. What is their populational average used? And the standard deviation? Please, could you explain how it was estimated?;

"Authors' response#5: Thank you so much. We calculated sample size by using a confidence interval approach for a single population proportion with 50% since no published study regarding prevalence of UP in the country settled up. Therefore, we used to assume a proportion of 50% to get the maximum sample to be representative."

Thank you for the explanation. I am afraid that their explanation wasn't enough to understand their sample size calculation. By the best of my limit knowledge, to calculate a sample by confidence interval approach for a single population proportion, you must know your standard deviation, not presented in the method area.

Also, if they assume a 50% margin of error, that is a chance of 50% that their null hypothesis is true. I think that the authors tried to use the sample calculation of other publications (Silwal M, Gurung, G., Shrestha, N., Gurung, A. and Ojha, S.: Prevalence and Factors Affecting Women with Uterine Prolapse in Lekhnath, Kaski, Nepal. , 9(2), pp.52-57. Journal of Gandaki Medical College-Nepal 2016, 9(2):6.). I suggest that they revise it with an epidemiologist or a mathematician.

Authors' response: Dear Reviewer, Thank you so much: to determine the maximum sample size for cross-sectional study; we determined sample size using a single population proportion formula with the assumption of the prevalence of UP among reproductive age 50%; p=0.5 and Z=1.96, Margin of error of 5% (d)= 0.05. we assumed a 5% of margin of error, not a 50%. A single population proportion gives a maximum sample size.

6- Material and methods: How were the randomization performed?;

"Authors' response#6: Thank you so much. From the total 28 kebeles in the Loma district, 30% of kebeles, i.e., a total of eight kebeles, were selected by lottery method. The assumption was to divide the total estimated sample size to the households of each kebele according to the proportion they contribute to the total study subjects. We allocated sample proportion for the selected eight kebeles based on sample size. Out of an estimated 422 participants, sample size was adjusted proportionally for the households of the selected eight kebeles. Then, households were selected by systematic random sampling method, i.e., selecting households at a fixed interval throughout a household list from a registration book of health post as provided by Health Extension Workers(HEWs) working at Health Post in each respective kebele."

The explanation of randomization must be described in the "Material and methods" area.

Authors' response: Dear Reviewer, Thank you so much: we revised how was the randomization performed in material and methods sction. Please see lines 125 -133. 

7- Material and methods: The authors describe that the patients were selected by an interview done by midwives? If they had symptoms, they were referred to the gynecologist to pelvic examination. The authors should describe as a weakness of the study that only symptomatic uterine prolapses were included in this study;

Authors' response#7: Thank you so much. We described the li mitation of the study in the limitation of the study section. Please see lines 234 to 235.

Done. I suggest that the authors present as a new paragraph only. It does not need that spotlight as a new subdivision in the discussion.

Authors' response: Dear Reviewer, Thank you so much: we revised it. Please see lines 256-257.

8- Material and methods: What questionnaire was used? What questions were used in each 3 main sections?;

"Authors' response#8: The interview question was composed of three main sections; the first two were phase 1 and the third was considered as phase two:

1) Socio-demographic variables and Obstetric and gynaecologic history (14 questions), 2) Questions regarding symptoms of uterine prolapsed (6questions),

3) The third section included two items; confirming by vaginal examination whether the women who reported symptomatic prolapse had anatomical prolapse or not, and the staging the prolapse was done then."

Thank you for the intent to answer the question. Unfortunately, the authors copy the explanation that was already in their article. The questions were not revealed in the article, neither in their review. It is not clear to readers what criteria was used to select patients. What questions were used? It is important to describe all the method used in the paper. Maybe their questions can be useful to others in the screening of UP.

Authors' response: Dear Reviewer, Thank you so much: the structured questionnaire used was included in the supplementary document section submitted with the manuscript. 

9- Material and methods: Was the questionnaire validated by a previous study? Please indicate in the text the reference;

"Authors' response#9 : Thanks so much; we are not sure that the questionnaire was not validated in the previous study."

The question 9 is related to question 8. They do not show the questions used. Also, they were not previous used in other articles. So, how they are so sure that their questions are adequate to the screening of their sample?

Authors' response: Dear Reviewer, Thank you so much: we reviewed different literature of similar studies. We checked the validity of the questionnaire using Cronbach alpha that was greater than 0.7 that indicates the questionnaire measures what it intended to measure. 

10- Material and methods (abstract): I would like to understand why authors included in the multivariate logistic regression the variables from the bivariate logistic regression with P < 0.25 instead of p<0.20. The authors should include this description in the Material and methods area of the main manuscript also;

"Authors' response #10: Thank you so much; We included variables with p< 0.25 for candidate variable for the final model to increase the chance of the variable became a significant predictor variable. We included the description in the material and methods section."

Thank you for the explanation. Unfortunately, their desire to make variables more prone to be significant biased their study. I suggest that it be analyze again, with the correct parameters often used in medical literature.

Authors' response: Dear Reviewer, Thank you so much: Biostaticians recommend using P < 0.25 as candidate variables in bivariate analysis for candidate variables for multivariate analysis. 

11- Results: I would like to understand the results presented: "This study identified that the prevalence of symptomatic and anatomical uterine prolapse was 6.6% (28) and 5.9% (25) respectively". How can be more symptomatic patients that actually anatomical identified uterine prolapse? The authors should explain;

"Authors' response #11: Dear reviewer, thank you so much, the data collectors were trained midwifes and creafully assessed symptomatic patients and most of the symptomatic patients identified as anatomically identified utrine prolapse by doctors."

Thank you, unfortunately the question remains: how they find more symptoms than they find the pathology? The symptomatology is not a diagnosis. So, in the initial screening with midwifes, there were 28 subjects that were suspicion of having UP. After doctor examination, only 25 were confirmed. They should make the corrections pointed.

Authors' response: Dear Reviewer, Thank you so much; in the initial screening with midwives, there were 28 subjects that were suspected of having UP. After the doctor examination, only 25 were confirmed.

12- Discussion: I suggest that the first paragraph, the authors describe their main objective findings;

"Authors' response #12: Dear reviewer, thank you so much, we included the statement "In this study; age at first marriage, a history of abortion, birth attendant who assisted the delivery, and place of delivery were independent factors associated with uterine prolapse."

Perfect done.

Authors' response: Dear Reviewer, Thank you so much

13- Abstract Conclusion: The authors may enhance their conclusion. What is the message? What they suggest to their public health as a possible action?;

"Authors' response #13: Dear reviewer, thank you so much. We enhance the conclusion, including the message: We recommend that the health system link primary health care to hospital-set for uterine prolapse treatment programs. Health institution delivery should be encouraged by the local government. Early marriage and unwanted pregnancy need to be prevented through appropriate strategies. Please See line 49 to 52."

Done.

Authors' response: Dear Reviewer, Thank you so much

14- Manuscript Conclusion: The authors should enhance the conclusion to better comprehension.

"Authors' response #14: Dear reviewer, thank you so much. We enhance the conclusion to better comprehension, including the "We recommend that the health system is linking primary health care to hospital-set for uterine prolapse treatment programs. Health institution delivery should be encouraged by the local government. Early marriage and unwanted pregnancy need to be prevented through appropriate strategies." Please see lines 240- 243."

Done.

Authors' response: Dear Reviewer, Thank you so much

15- Introduction: They erased lines 118-119: "According to a projection from the 2007 national census, Ethiopia had nearly 110 million 119 inhabitants in 2019, with 23.4% of women of reproductive age.". The subsequent phase must be adjusted: "Given this massive potentially…"

Authors' response: Dear Reviewer, Thank you so much; We revised it. Please see line 97.

16- Discussion, lines 250-251: Symptomatic prolapse in your sample wasn't 6.6%. If the patients had gynecological examination by doctors and they diagnosticated only 25 of the total sample of 422, your result is 5.9%. There are 3 patients screened to UP that had not the diagnosis of UP made.

Authors' response: Dear Reviewer, Thank you so much; we revised it

17- Figure 2: The same as the previous: the presence of uterine prolapse was only in 25 patients (5.9%). The authors should revise it. A questionary cannot be the gold standard to UP diagnosis. The clinical evaluation is the gold standard.

Authors' response: Dear Reviewer, Thank you so much; we revised it

18- Table 1: marital status: Is missing data of 1 patient. They should revise.

Authors' response: Dear Reviewer, Thank you so much; we revised it

19- Table 1: religion: Is missing data of 1.7% of their sample. They should revise.

Authors' response: Dear Reviewer, Thank you so much; we revised it

---

## [Decision Letter · Decision Letter 2]

22 Sep 2021

PONE-D-20-40659R2Uterine prolapse and associated factors among reproductive-age women in south-west Ethiopia : a community-based cross-sectional studyPLOS ONE

Dear Dr. badacho,

Thank you for submitting your manuscript to PLOS ONE. After careful consideration, we feel that it has merit but does not fully meet PLOS ONE’s publication criteria as it currently stands. Therefore, we invite you to submit a revised version of the manuscript that addresses the points raised during the review process. Please submit your revised manuscript by Nov 06 2021 11:59PM. If you will need more time than this to complete your revisions, please reply to this message or contact the journal office at plosone@plos.org. Please include the following items when submitting your revised manuscript:A rebuttal letter that responds to each point raised by the academic editor and reviewer(s). You should upload this letter as a separate file labeled 'Response to Reviewers'.A marked-up copy of your manuscript that highlights changes made to the original version. You should upload this as a separate file labeled 'Revised Manuscript with Track Changes'.An unmarked version of your revised paper without tracked changes. You should upload this as a separate file labeled 'Manuscript'.

We look forward to receiving your revised manuscript.

Kind regards,

Richard Kao Lee, M.D.

Academic Editor

PLOS ONE

Reviewers' comments:

Reviewer's Responses to Questions

**Comments to the Author**

1. If the authors have adequately addressed your comments raised in a previous round of review and you feel that this manuscript is now acceptable for publication, you may indicate that here to bypass the “Comments to the Author” section, enter your conflict of interest statement in the “Confidential to Editor” section, and submit your "Accept" recommendation.

Reviewer #1: All comments have been addressed

Reviewer #3: (No Response)

2. Is the manuscript technically sound, and do the data support the conclusions?

Reviewer #1: Yes

Reviewer #3: Partly

3. Has the statistical analysis been performed appropriately and rigorously? 

Reviewer #1: Yes

Reviewer #3: No

4. Have the authors made all data underlying the findings in their manuscript fully available?

Reviewer #1: Yes

Reviewer #3: Yes

5. Is the manuscript presented in an intelligible fashion and written in standard English?

Reviewer #1: Yes

Reviewer #3: No

6. Review Comments to the Author

Reviewer #1: In my opinion can be accept, the paper is very good and also the time related to peer review is too much and I think to make a decision in short time no 1 year

Reviewer #3: About the revised article “Uterine prolapse and associated factors among reproductive age women in southwest Ethiopia”: The authors’ aim was to evaluate the prevalence and factors associated with uterine prolapse among women of reproductive age throw a cross-sectional study. The end point was to analyze risk factors associated with uterine prolapse.

First, I would like to congratulate the authors by their effort in re-reviewing the reviewers’ statements. Each point with the revision needed (in my humble opinion):

1- The article needs a native English speaker review; ---- done

2- The authors should describe the full words in first passage abbreviations (i.e. AOR, COR);

“Authors' response#2: Thank you so much. We fully desrcibed the full words in first passage abbreviation. Please see line 49.”

------ Done

3- Introduction: The introduction is too long. The authors should shrink it. I suggest that the treatment section can be suppressed; - done partially. The introduction remains too long. It would be better that they cut unnecessary parts. Authors' response: Dear Reviewer, Thank you so much: we revised the introduction part.

------ Done partially. The introduction remains too long. It would be better that they cut unnecessary parts.

4- Introduction: last paragraph: I am not sure that they present the prevalence with this cross-sectional study. I understood that they analyzed a few sample of their population. How they make to this sample be representative? They should revise it; “Authors' response#4: Thank you so much; we determined sample size based on scientific method size determination which gave us enough sample size. Participants were randomly selected in the study.” Thank you, but the answer is still lacking to me, as you can see the next question. Also, the authors described that the screening was done by trained midwifes, finding 28 subjects that were suspicion of having UP. After doctor examination, only 25 confirmed. Also, the authors didn’t comment, but not even all UP the patients presents with symptoms. So, they can only estimate the incidence of symptomatic UP. The prevalence of UP is still not known, because only a few subjects were examined. Authors' response: Dear Reviewer, Thank you so much: even though we determined sample size using the scientific method of sample size calculation the subjected examined for UP was few that affected the prevalence of UP among the reproductive age group.

--- Thank you, but the answer is still lacking to me. The answer has not clarified my previous questions. Also, the authors described that the screening was done by trained midwifes, finding 28 subjects that were suspicion of having UP. After doctor examination, only 25 confirmed. Also, the authors didn’t comment, but not even all UP the patients presents with symptoms. So, they can only estimate the incidence of symptomatic UP. The prevalence of UP is still not known, because only a few subjects were examined. Unfortunately, they do not know how many asymptomatic reproductive women have UP, so it is possible that their population prevalence be higher than they calculated;

5- Material and methods: “The sample size was calculated by using a single population formula with p=0.5 margin of error and (d) = 0.05 ”. I couldn’t understand how the authors calculated their sample size. What is their populational average used? And the standard deviation? Please, could you explain how it was estimated?; “Authors' response#5: Thank you so much. We calculated sample size by using a confidence interval approach for a single population proportion with 50% since no published study regarding prevalence of UP in the country settled up. Therefore, we used to assume a proportion of 50% to get the maximum sample to be representative.” Thank you for the explanation. I am afraid that their explanation wasn’t enough to understand their sample size calculation. By the best of my limit knowledge, to calculate a sample by confidence interval approach for a single population proportion, you must know your standard deviation, not presented in the method area. Also, if they assume a 50% margin of error, that is a chance of 50% that their null hypothesis is true. I think that the authors tried to use the sample calculation of other publication (Silwal M, Gurung, G., Shrestha, N., Gurung, A. and Ojha, S.: Prevalence and Factors Affecting Women with Uterine Prolapse in Lekhnath, Kaski, Nepal. , 9(2), pp.52-57. Journal of Gandaki Medical College-Nepal 2016, 9(2):6.). I suggest that they revise it with and epidemiologist or a mathematician. Authors' response: Dear Reviewer, Thank you so much: to determine the maximum sample size for cross-sectional study; we determined sample size using a single population proportion formula with the assumption of the prevalence of UP among reproductive age 50%; p=0.5 and Z=1.96, Margin of error of 5% (d)= 0.05. we assumed a 5% of margin of error, not a 50%. A single population proportion gives a maximum sample size.

--------- Thank you. First I think you will have to correct the information in the material and methods. You answered here “Margin of error of 5% (d)= 0.05”; however, in the article you wrote “The sample size was calculated by using a single population formula with p=0.5 margin of error and (d) = 0.05 ”. Maybe your desire was to write: “The sample size was calculated by using a single population formula with a proportion of 0.5 and a margin of error of (d) = 0.05 ”

You assume a prevalence of 50%. You found around 6% of UP in your sample. You have estimated a high prevalence of UP compared with your sample results (6%). By the methodology used, your article was underpower. Also, there was already an evidence in Ethiopia of 22.3% (Silwal M, Gurung, G., Shrestha, N., Gurung, A. and Ojha, S.: Prevalence and Factors Affecting Women with Uterine Prolapse in Lekhnath, Kaski, Nepal. , 9(2), pp.52-57. Journal of Gandaki Medical College-Nepal 2016, 9(2):6.) that you could used to estimate your sample size.

If we consider that your sample calculation was adequate (to 6% of prevalence), your result is not a prevalence of 6%, because you have not diagnosticated women with asymptomatic UP. In my interpretation, you can conclude that the incidence of symptomatic UP of your sample was 5.9%.

6- Material and methods: How were the randomization performed?; “Authors' response#6: Thank you so much. From the total 28 kebeles in the Loma district, 30% of kebeles, i.e., a total of eight kebeles, were selected by lottery method. The assumption was to divide the total estimated sample size to the households of each kebele according to the proportion they contribute to the total study subjects. We allocated sample proportion for the selected eight kebeles based on sample size. Out of an estimated 422 participants, sample size was adjusted proportionally for the households of the selected eight kebeles. Then, households were selected by systematic random sampling method, i.e., selecting households at a fixed interval throughout a household list from a registration book of health post as provided by Health Extension Workers(HEWs) working at Health Post in each respective kebele.” The explanation of randomization must be described in the “Material and methods” area. Authors' response: Dear Reviewer, Thank you so much: we revised how was the randomization performed in material and methods sction. Please see lines 125 -133.

---- Thank you. If I understood right, you performed a simple random sampling, that is correct? If so, I suggest that you can say that you raffle 8 areas of Loma district and after that selected the sample by a simple random sampling in these areas. It will be less confusing to readers. I had to study what is a kebele. Thank you for the knowledge.

7- Material and methods: The authors describe that the patients were selected by an interview done by midwives? If they had symptoms, they were referred to the gynecologist to pelvic examination. The authors should describe as a weakness of the study that only symptomatic uterine prolapses were included in this study; Authors' response#7: Thank you so much. We described the limitation of the study in the limitation of the study section. Please see lines 234 to 235. Done. I suggest that the authors present as a new paragraph only. It does not need that spotlight as a new subdivision in the discussion. Authors' response: Dear Reviewer, Thank you so much: we revised it. Please see lines 256-257.

---- Thank you. The authors could present a paragraph with strengths and limitations of their work. They made only a final phrase of 1 limitation. They can cite that the article sample is underpower also, because of their prior prevalence estimation was more than 8 times higher than what they found.

8- Material and methods: What questionnaire were used? What questions were used in each 3 main sections?; “Authors' response#8: The interview question was composed of three main sections; the first two were phase 1 and the third was considered as phase two:

1) Socio-demographic variables and Obstetric and gynaecologic history (14 questions), 2) Questions regarding symptoms of uterine prolapsed (6questions),

3) The third section included two items; confirming by vaginal examination whether the women who reported symptomatic prolapse had anatomical prolapse or not, and the staging the prolapse was done then.” Thank you for the intent to answer the question. Unfortunately, the authors copy the explanation that was already in their article. The questions were not revealed in the article, neither in their review. It is not clear to readers what criteria was used to select patients. What questions were used? It is important to describe all the method used in the paper. Maybe their questions can be useful to others in the screening of UP. Authors' response: Dear Reviewer, Thank you so much: the structured questionnaire used was included in the supplementary document section submitted with the manuscript.

-------- Thank you, I could understand the questionary used. Is there any reference of its use as a screening method to UP? Is it the first time used? If it is the first time used, I suggest that they validate it as a tool to population screening. In this case, all the sample must have gynecologic examination to rightly diagnose UP. The use of a non-validated questionary reinforces the question 5 (they could not conclude they population prevalence).

9- Material and methods: Were the questionnaire validated by a previous study? Please indicate in the text the reference; “Authors' response#9 : Thanks so much; we are not sure that the questionnaire was not validated in the previous study.” The question 9 is related to question 8. They do not show questions used. Also, they were not previous used in other articles. So, how they are so sure that their questions are adequate to screening of their sample? Authors' response: Dear Reviewer, Thank you so much: we reviewed different literature of similar studies. We checked the validity of the questionnaire using Cronbach alpha that was greater than 0.7 that indicates the questionnaire measures what it intended to measure.

-------- Thank you. The Cronbach alpha is used to as a measure of internal consistency of a questionary, it is not used as a validation tool. That is OK, you already answer that the text was not previously validated in the question 8. Who were the specialists that participate in the Cronbach alpha calculation? Were the authors of this article?

10- Material and methods (abstract): I would like to understand why authors included in the multivariate logistic regression the variables from the bivariate logistic regression with P < 0.25 instead of p<0.20. The authors should include this description in the Material and methods area of the main manuscript also;

“Authors' response #10: Thank you so much; We included variables with p< 0.25 for candidate variable for the final model to increase the chance of the variable became a significant predictor variable. We included the description in the material and methods section.”

Thank you for the explanation. Unfortunately, their desire to make variables more prone to be significant biased their study. I suggest that it be analyze again, with the correct parameters often used in medical literature.

Authors' response: Dear Reviewer, Thank you so much: Biostaticians recommend using P < 0.25 as candidate variables in bivariate analysis for candidate variables for multivariate analysis.

--------- Thank you for your response. I am still confuse. You describe on the abstract’s methodology area that you used variables with P-values less than 0.25 in bivariate logistic regression were further examined using multivariate logistic regression to investigate associations between the dependent variable and independent variables. However, on the methology’s area of the complete article, you do not describe it. Also, your tables did not show what variables was calculated with p<0.25 to use in the multivariable analysis. The authors should clarify or rectify the information.

11- Results: I would like to understand the results presented: “This study identified that the prevalence of symptomatic and anatomical uterine prolapse was 6.6% (28) and 5.9% (25) respectively”. How can be more symptomatic patients that actually anatomical identified uterine prolapse? The authors should explain;

“Authors' response #11: Dear reviewer, thank you so much, the data collectors were trained midwifes and creafully assessed symptomatic patients and most of the symptomatic patients identified as anatomically identified utrine prolapse by doctors.”

Thank you, unfortunately the question remains: how they find more symptoms than they find the pathology? The symptomatology is not a diagnosis. So, in the initial screening with midwifes, there were 28 subjects that were suspicion of having UP. After doctor examination, only 25 confirmed. They should make the corrections pointed.

Authors' response: Dear Reviewer, Thank you so much; in the initial screening with midwives, there were 28 subjects that were suspected of having UP. After the doctor examination, only 25 were confirmed.

------- Thank you, I understood exactly that. So you agree that only 25 were confirmed as having UP. In that way, you cannot say that there were 28 patients with symptomatic UP. You must rectify the information in the text. Also, you cannot affirm that the prevalence of UP in your sample was 5.9%, because you do not know that (there are asymptomatic UP that were not included in your article).

12- Discussion: I suggest that the first paragraph, the authors describe their main objective findings;

“Authors' response #12: Dear reviewer, thank you so much, we included the statement “In this study; age at first marriage, a history of abortion, birth attendant who assisted the delivery, and place of delivery were independent factors associated with uterine prolapse.”

------- Perfect done.

13- Abstract Conclusion: The authors may enhance their conclusion. What is the message? What they suggest to their public health as a possible action?;

“Authors' response #13: Dear reviewer, thank you so much. We enhance the conclusion, including the message: We recommend that the health system link primary health care to hospital-set for uterine prolapse treatment programs. Health institution delivery should be encouraged by the local government. Early marriage and unwanted pregnancy need to be prevented through appropriate strategies. Please See line 49 to 52.”

------ Done.

14- Manuscript Conclusion: The authors should enhance the conclusion to better comprehension.

“Authors' response #14: Dear reviewer, thank you so much. We enhance the conclusion to better comprehension, including the "We recommend that the health system is linking primary health care to hospital-set for uterine prolapse treatment programs. Health institution delivery should be encouraged by the local government. Early marriage and unwanted pregnancy need to be prevented through appropriate strategies." Please see lines 240- 243.”

------- Done.

15- Introduction: They erased lines 118-119: “According to a projection from the 2007 national census, Ethiopia had nearly 110 million 119 inhabitants in 2019, with 23.4% of women of reproductive age.”. The subsequent phase must be adjusted: “Given this massive potentially…”

Authors' response: Dear Reviewer, Thank you so much; We revised it. Please see line 97.

------- Thank you. You have cutted the passage “According to a projection from the 2007 national census, Ethiopia had nearly 110 million 119 inhabitants in 2019, with 23.4% of women of reproductive age.” However, the subsequent phrase has lost its meaning. The authors should revise it.

16- Discussion, lines 250-251: Symptomatic prolapse in your sample wasn’t 6.6%. If the patients had gynecological examination by doctors and they diagnosticated only 25 of the total sample of 422, your result is 5.9%. There are 3 patients screened to UP that had not the diagnosis of UP made.

Authors' response: Dear Reviewer, Thank you so much; we revised it

------- Thank you. I already commented in the previous questions. Unfortunately, you have not reviewed.

17- Figure 2: The same as the previous: the presence of uterine prolapse was only in 25 patients (5.9%). The authors should revise it. A questionary cannot be the gold standard to UP diagnosis. The clinical evaluation is the gold standard.

Authors' response: Dear Reviewer, Thank you so much; we revised it

------- Thank you. I already commented in the previous questions. Unfortunately, you have not reviewed.

18- Table 1: marital status: Is missing data of 1 patient. They should revise.

Authors' response: Dear Reviewer, Thank you so much; we revised it

------ Done partially. The percentage is missing 0.2%

19- Table 1: religion: Is missing data of 1.7% of their sample. They should revise.

Authors' response: Dear Reviewer, Thank you so much; we revised it

------ Done

20 – Abstract, “The objective of this study was to assess the prevalence of and factors associated with uterine prolapse in women of reproductive age in Ethiopia.”: The objective was not accomplished by this study. UP can be asymptomatic. So, without a medical evaluation, it is not possible to determine the prevalence of UP in their population. The inclusion criteria were done by non-medical interview (using a questionary). The authors should revise it.

21 – Abstract, results, “The prevalence of symptomatic and anatomical uterine prolapse was 6.6% (28) and 5.9% (25), respectively”: Sorry, but the authors already said that 3 patients of the 28 symptomatic did not confirm the hypothesis of UP after medical examination. They did not revise the manuscript as previously indicated in question 04.

22 – Introduction, “UP is the most common gynecological health problem contributing to maternal morbidity and mortality in women of reproductive age in developing countries”: The authors should show the reference of their statement.

23 – Introduction: It is still too long. They should shrink it.

24 - The authors should describe the full words in first passage abbreviations (i.e. COR);

25 – The authors indexed the previous and the new revision, it was a little bit confuse to identification of the new one.

Cordially

7. PLOS authors have the option to publish the peer review history of their article (what does this mean?). If published, this will include your full peer review and any attached files.

Reviewer #1: No

Reviewer #3: **Yes: **Ricardo Pedrini Cruz

---

## [Author Response · Author response to Decision Letter 2]

23 Sep 2021

PONE-D-20-40659R2

Uterine prolapse and associated factors among reproductive-age women in Dawro Zone, southwest Ethiopia: a community based cross-sectional study

PLOS ONE

Subject: Point by point response to reviewers’ comments

 Dear PLOS ONE Editorial Manager,

Please kindly find the submitted revised manuscript with ID number PONE-D-20-40659R2, entitled “Uterine prolapse and associated factors among reproductive-age women in Dawro Zone, southwest Ethiopia: a community based cross-sectional study” point to point response to the reviewer’s comment for kindly consideration. 

With kind regards,

Abebe Sorsa (Corresponding Author)

Authors point by point response to Reviewer’s Responses to Questions

Reviewer’s Responses to Questions

Comments to the Author

1. If the authors have adequately addressed your comments raised in a previous round of review and you feel that this manuscript is now acceptable for publication, you may indicate that here to bypass the “Comments to the Author” section, enter your conflict of interest statement in the “Confidential to Editor” section, and submit your “Accept” recommendation.

Reviewer #1: All comments have been addressed

Authors’ response: Dear Reviewer, Thank you so much

Reviewer #3: (No Response)

2. Is the manuscript technically sound, and do the data support the conclusions?

Reviewer #1: Yes

Authors’ response: Dear Reviewer, Thank you so much

Reviewer #3: Partly

Authors’ response: Dear Reviewer, Thank you so much. We have revised the manuscript 

3. Has the statistical analysis been performed appropriately and rigorously?

Reviewer #1: Yes

Authors’ response: Dear Reviewer, Thank you so much

Reviewer #3: No

Authors’ response: Dear Reviewer, Thank you so much; we have revised the manuscript 

4. Have the authors made all data underlying the findings in their manuscript fully available?

Reviewer #1: Yes

Authors’ response: Dear Reviewer, Thank you so much

Reviewer #3: Yes

Authors’ response: Dear Reviewer, Thank you so much 

5. Is the manuscript presented in an intelligible fashion and written in standard English?

Reviewer #1: Yes

Authors’ response: Dear Reviewer, Thank you so much,

Reviewer #3: No

Authors’ response: Dear Reviewer, Thank you so much; we utilized the nextgen professional editing for this manuscript and used Grammarly for checking it. 

6. Review Comments to the author

Reviewer #1: In my opinion can be accepted, the paper is very good and also the time related to peer review is too much, and I think to make a decision in a short time, no one year.

Authors’ response: Dear Reviewer, Thank you so much for your time and reviewing this manuscript

Reviewer #3: About the revised article “Uterine prolapse and associated factors among reproductive-age women in southwest Ethiopia”: The authors’ aim was to evaluate the prevalence and factors associated with uterine prolapse among women of reproductive age throw a cross-sectional study. The endpoint was to analyze risk factors associated with uterine prolapse.

First, I would like to congratulate the authors by their effort in re-reviewing the reviewers’ statements. Each point with the revision needed (in my humble opinion):

Authors’ response: Dear Reviewer, Thank you so much for your time and reviewing this manuscript

1- The article needs a native English speaker review; ---- done

Authors’ response: Dear Reviewer, Thank you so much for time and reviewing this manucript

2- The authors should describe the full words in first passage abbreviations (i.e. AOR, COR);

“Authors’ response#2: Thank you so much. We fully desrcibed the full words in first passage abbreviation. Please see line 49.”------ Done.

Authors’ response: Dear Reviewer, Thank you so much for time and reviewing this manucript

3- Introduction: The introduction is too long. The authors should shrink it. I suggest that the treatment section can be suppressed; - done partially. The introduction remains too long. It would be better that they cut unnecessary parts. Authors’ response: Dear Reviewer, Thank you so much: we revised the introduction part.------ Done partially. The introduction remains too long. It would be better that they cut unnecessary parts.

Authors’ response: Dear Reviewer, Thank you so much; we revised the introduction part. 

4- Introduction: last paragraph: I am not sure that they present the prevalence with this cross-sectional study. I understood that they analyzed a few sample of their population. How they make to this sample be representative? They should revise it; “Authors’ response#4: Thank you so much; we determined sample size based on scientific method size determination which gave us enough sample size. Participants were randomly selected in the study.” Thank you, but the answer is still lacking to me, as you can see the next question. Also, the authors described that the screening was done by trained midwives, finding 28 subjects that were suspicion of having UP. After doctor examination, only 25 confirmed. Also, the authors didn’t comment, but not even all UP the patients presents with symptoms. So, they can only estimate the incidence of symptomatic UP. The prevalence of UP is still not known, because only a few subjects were examined. Authors’ response: Dear Reviewer, Thank you so much: even though we determined sample size using the scientific method of sample size calculation the subjected examined for UP was few that affected the prevalence of UP among the reproductive age group.

--- Thank you, but the answer is still lacking to me. The answer has not clarified my previous questions. Also, the authors described that the screening was done by trained midwifes, finding 28 subjects that were suspicion of having UP. After doctor examination, only 25 confirmed. Also, the authors didn’t comment, but not even all UP the patients presents with symptoms. So, they can only estimate the incidence of symptomatic UP. The prevalence of UP is still not known, because only a few subjects were examined. Unfortunately, they do not know how many asymptomatic reproductive women have UP, so it is possible that their population prevalence be higher than they calculated;

Authors’ response: Dear Reviewer, Thank you so much. We estimated the prevalence of UP among the general reproductive age population of the Loma district. Even though the sample size was small, we determined the sample size using scientific methods. 

5- Material and methods: “The sample size was calculated by using a single population formula with p=0.5 margin of error and (d) = 0.05 ”. I couldn’t understand how the authors calculated their sample size. What is their populational average used? And the standard deviation? Please, could you explain how it was estimated?; “Authors’ response#5: Thank you so much. We calculated sample size using a confidence interval approach for a single population proportion with 50% since no published study regarding the prevalence of UP in the country settled up. Therefore, we used to assume a proportion of 50% to get the maximum sample to be representative.” Thank you for the explanation. I am afraid that their explanation wasn’t enough to understand their sample size calculation. By the best of my limit knowledge, to calculate a sample by confidence interval approach for a single population proportion, you must know your standard deviation, not presented in the method area. Also, if they assume a 50% margin of error, that is a chance of 50% that their null hypothesis is true. The authors tried to use the sample calculation of other publications (Silwal M, Gurung, G., Shrestha, N., Gurung, A. and Ojha, S.: Prevalence and Factors Affecting Women with Uterine Prolapse in Lekhnath, Kaski, Nepal. , 9(2), pp.52-57. Journal of Gandaki Medical College-Nepal 2016, 9(2):6.). I suggest that they revise it with and epidemiologist or a mathematician. Authors’ response: Dear Reviewer, Thank you so much: to determine the maximum sample size for cross-sectional study; we determined sample size using a single population proportion formula with the assumption of the prevalence of UP among reproductive age 50%; p=0.5 and Z=1.96, Margin of error of 5% (d)= 0.05. we assumed a 5% of margin of error, not a 50%. A single population proportion gives a maximum sample size.

--------- Thank you. First I think you will have to correct the information in the material and methods. You answered here “Margin of error of 5% (d)= 0.05”; however, in the article you wrote “The sample size was calculated by using a single population formula with p=0.5 margin of error and (d) = 0.05 ”. Maybe your desire was to write: “The sample size was calculated by using a single population formula with a proportion of 0.5 and a margin of error of (d) = 0.05 ”

You assume a prevalence of 50%. You found around 6% of UP in your sample. You have estimated a high prevalence of UP compared with your sample results (6%). By the methodology used, your article was underpower. Also, there was already an evidence in Ethiopia of 22.3% (Silwal M, Gurung, G., Shrestha, N., Gurung, A. and Ojha, S.: Prevalence and Factors Affecting Women with Uterine Prolapse in Lekhnath, Kaski, Nepal. , 9(2), pp.52-57. Journal of Gandaki Medical College-Nepal 2016, 9(2):6.) that you could used to estimate your sample size.

If we consider that your sample calculation was adequate (to 6% of prevalence), your result is not a prevalence of 6%, because you have not diagnosticated women with asymptomatic UP. In my interpretation, you can conclude that the incidence of symptomatic UP of your sample was 5.9%.

Authors’ response: Dear Reviewer, Thank you so much for your time and reviewing this manuscript, perfectly we desired to write, “The sample size was calculated by using a single population formula with a proportion of 0.5 and a margin of error of (d) = 0.05 ”. we revised it. We used a proportion of 0.5 to get the maximum sample size. Suppose we would use prevalence UP evidence in Ethiopia of 22.3%, giving us a smaller sample size comparing using a proportion of 0.5. We intended to include a maximum sample size for our study even though the underpower. 

6- Material and methods: How were the randomization performed?; “Authors’ response#6: Thank you so much. From the total 28 kebeles in the Loma district, 30% of kebeles, i.e., a total of eight kebeles, were selected by the lottery method. The assumption was to divide the total estimated sample size to the households of each kebele according to the proportion they contribute to the total study subjects. We allocated sample proportions for the selected eight kebeles based on sample size. Out of an estimated 422 participants, the sample size was adjusted proportionally for the households of the selected eight kebeles. Then, households were selected by systematic random sampling method, i.e., selecting households at a fixed interval throughout a household list from a registration book of health post as provided by Health Extension Workers(HEWs) working at Health Post in each respective kebele.” The explanation of randomization must be described in the “Material and methods” area. Authors’ response: Dear Reviewer, Thank you so much: we revised how was the randomization performed in material and methods sction. Please see lines 125 -133.

---- Thank you. If I understood right, you performed a simple random sampling, that is correct? If so, I suggest that you can say that you raffle 8 areas of Loma district and after that selected the sample by a simple random sampling in these areas. It will be less confusing to readers. I had to study what is a kebele. Thank you for the knowledge.

Authors’ response: Dear Reviewer, Thank you so much for your time: we revised it accordingly. Kebele is the lowest administrative unit in Ethiopia. 

7- Material and methods: The authors describe that the patients were selected by an interview done by midwives? If they had symptoms, they were referred to the gynecologist to pelvic examination. The authors should describe as a weakness of the study that only symptomatic uterine prolapses were included in this study; Authors’ response#7: Thank you so much. We described the limitation of the study in the limitation of the study section. Please see lines 234 to 235. Done. I suggest that the authors present as a new paragraph only. It does not need that spotlight as a new subdivision in the discussion. Authors’ response: Dear Reviewer, Thank you so much: we revised it. Please see lines 256-257.

---- Thank you. The authors could present a paragraph with strengths and limitations of their work. They made only a final phrase of 1 limitation. They can cite that the article sample is underpower also, because of their prior prevalence estimation was more than 8 times higher than what they found.

Authors’ response#7: Dear Reviewer, Thank you so much for your time, we revised manuscript accordingly. Please see lines 238 – 240.

8- Material and methods: What questionnaires were used? What questions were used in each 3 main sections?; “Authors’ response#8: The interview question was composed of three main sections; the first two were phase 1 and the third was considered as phase two:

1) Socio-demographic variables and Obstetric and gynaecologic history (14 questions), 2) Questions regarding symptoms of uterine prolapsed (6questions),

3) The third section included two items; confirming by vaginal examination whether the women who reported symptomatic prolapse had anatomical prolapse or not, and the staging the prolapse was done then.” Thank you for the intent to answer the question. Unfortunately, the authors copy the explanation that was already in their article. The questions were not revealed in the article, neither in their review. It is not clear to readers what criteria was used to select patients. What questions were used? It is important to describe all the method used in the paper. Maybe their questions can be useful to others in the screening of UP. Authors’ response: Dear Reviewer, Thank you so much: the structured questionnaire used was included in the supplementary document section submitted with the manuscript.

-------- Thank you, I could understand the questionary used. Is there any reference of its use as a screening method to UP? Is it the first time used? If it is the first time used, I suggest that they validate it as a tool to population screening. In this case, all the sample must have gynecologic examination to rightly diagnose UP. The use of a non-validated questionary reinforces the question 5 (they could not conclude they population prevalence).

Authors’ response#8: Dear Reviewer, Thank you so much for your time. We reviewed different works of literature on similar studies and adopted and constructed the tool to screen UP. 

9- Material and methods: Were the questionnaire validated by a previous study? Please indicate in the text the reference; “Authors’ response#9 : Thanks so much; we are not sure that the questionnaire was not validated in the previous study.” The question 9 is related to question 8. They do not show questions used. Also, they were not previous used in other articles. So, how they are so sure that their questions are adequate to screening of their sample? Authors’ response: Dear Reviewer, Thank you so much: we reviewed different literature of similar studies. We checked the validity of the questionnaire using Cronbach alpha that was greater than 0.7 that indicates the questionnaire measures what it intended to measure.

-------- Thank you. The Cronbach alpha is used to as a measure of internal consistency of a questionary, it is not used as a validation tool. That is OK, you already answered that the text was not previously validated in question 8. Who were the specialists that participated in the Cronbach alpha calculation? Were the authors of this article?

Authors’ response#9: Dear Reviewer, Thank you so much for your time. We reviewed different works of literature on similar studies and adopted and constructed the tool to screen UP. Pretest done ensure the reliability of the study tools and expert rating of the study tool done to ensure content validity of the tool by measuring the content validity index (CVI). Gynaecology experts, researchers were rated for the validity of the tool. 

10- Material and methods (abstract): I would like to understand why the authors included in the multivariate logistic regression the variables from the bivariate logistic regression with P < 0.25 instead of p<0.20. The authors should include this description in the Material and methods area of the main manuscript also;

“Authors’ response #10: Thank you so much; We included variables with p< 0.25 for candidate variable for the final model to increase the chance of the variable became a significant predictor variable. We included the description in the material and methods section.”

Thank you for the explanation. Unfortunately, their desire to make variables more prone to be significant biased their study. I suggest that it be analyze again, with the correct parameters often used in medical literature.

Authors’ response: Dear Reviewer, Thank you so much: Biostaticians recommend using P < 0.25 as candidate variables in bivariate analysis for candidate variables for multivariate analysis.

--------- Thank you for your response. I am still confused. You describe on the abstract’s methodology area that you used variables with P-values less than 0.25 in bivariate logistic regression were further examined using multivariate logistic regression to investigate associations between the dependent variable and independent variables. However, in the methology’s area of the complete article, you do not describe it. Also, your tables did not show what variables was calculated with p<0.25 to use in the multivariable analysis. The authors should clarify or rectify the information.

Authors’ response#10: Dear Reviewer, Thank you so much for your time. We revised accordingly.

11- Results: I would like to understand the results presented: “This study identified that the prevalence of symptomatic and anatomical uterine prolapse was 6.6% (28) and 5.9% (25) respectively”. How can be more symptomatic patients that actually anatomical identified uterine prolapse? The authors should explain;

“Authors’ response #11: Dear reviewer, thank you so much, the data collectors were trained midwifes and creafully assessed symptomatic patients and most of the symptomatic patients identified as anatomically identified utrine prolapse by doctors.”

Thank you, unfortunately the question remains: how they find more symptoms than they find the pathology? The symptomatology is not a diagnosis. So, in the initial screening with midwifes, there were 28 subjects that were suspicion of having UP. After doctor examination, only 25 confirmed. They should make the corrections pointed.

Authors’ response: Dear Reviewer, Thank you so much; in the initial screening with midwives, there were 28 subjects that were suspected of having UP. After the doctor examination, only 25 were confirmed.

------- Thank you, I understood exactly that. So you agree that only 25 were confirmed as having UP. In that way, you cannot say that there were 28 patients with symptomatic UP. You must rectify the information in the text. Also, you cannot affirm that the prevalence of UP in your sample was 5.9%, because you do not know that (there are asymptomatic UP that were not included in your article).

Authors’ response#11: Dear Reviewer, Thank you so much for your time. We revised accordingly.

12- Discussion: I suggest that in the first paragraph, the authors describe their main objective findings;

“Authors’ response #12: Dear reviewer, thank you so much, we included the statement “In this study; age at first marriage, a history of abortion, birth attendant who assisted the delivery, and place of delivery were independent factors associated with uterine prolapse.”

------- Perfect done.

Authors’ response#12: Dear Reviewer, Thank you so much for your time.

13- Abstract Conclusion: The authors may enhance their conclusion. What is the message? What they suggest to their public health as a possible action?;

“Authors’ response #13: Dear reviewer, thank you so much. We enhance the conclusion, including the message: We recommend that the health system link primary health care to hospital-set for uterine prolapse treatment programs. Health institution delivery should be encouraged by the local government. Early marriage and unwanted pregnancy need to be prevented through appropriate strategies. Please See line 49 to 52.”

------ Done.

Authors’ response#13: Dear Reviewer, Thank you so much for your time.

14- Manuscript Conclusion: The authors should enhance the conclusion to better comprehension.

“Authors’ response #14: Dear reviewer, thank you so much. We enhance the conclusion to better comprehension, including the “We recommend that the health system is linking primary health care to hospital-set for uterine prolapse treatment programs. Health institution delivery should be encouraged by the local government. Early marriage and unwanted pregnancy need to be prevented through appropriate strategies.” Please see lines 240- 243.”

------- Done.

Authors’ response#14: Dear Reviewer, Thank you so much for your time.

15- Introduction: They erased lines 118-119: “According to a projection from the 2007 national census, Ethiopia had nearly 110 million 119 inhabitants in 2019, with 23.4% of women of reproductive age.”. The subsequent phase must be adjusted: “Given this massive potentially…”

Authors’ response: Dear Reviewer, Thank you so much; We revised it. Please see line 97.

------- Thank you. You have cutted the passage “According to a projection from the 2007 national census, Ethiopia had nearly 110 million 119 inhabitants in 2019, with 23.4% of women of reproductive age.” However, the subsequent phrase has lost its meaning. The authors should revise it.

Authors’ response#15: Dear Reviewer, Thank you so much for your time. We revised accordingly.

16- Discussion, lines 250-251: Symptomatic prolapse in your sample wasn’t 6.6%. If the patients had gynecological examination by doctors and they diagnosticated only 25 of the total sample of 422, your result is 5.9%. There are 3 patients screened to UP that had not the diagnosis of UP made.

Authors’ response: Dear Reviewer, Thank you so much; we revised it

------- Thank you. I already commented in the previous questions. Unfortunately, you have not reviewed.

Authors’ response #16: Dear Reviewer, Thank you so much for your time. We revised accordingly.Please see lines 219-220.

17- Figure 2: The same as the previous: the presence of uterine prolapse was only in 25 patients (5.9%). The authors should revise it. A questionary cannot be the gold standard to UP diagnosis. The clinical evaluation is the gold standard.

Authors’ response: Dear Reviewer, Thank you so much; we revised it

------- Thank you. I already commented in the previous questions. Unfortunately, you have not reviewed.

Authors’ response#17: Dear Reviewer, Thank you so much for your time. We revised accordingly.Please see figure 2.

18- Table 1: marital status: Is missing data of 1 patient. They should revise.

Authors’ response: Dear Reviewer, Thank you so much; we revised it

------ Done partially. The percentage is missing 0.2%

Authors’ response#18: Dear Reviewer, Thank you so much for your time. We revised

19- Table 1: religion: Is missing data of 1.7% of their sample. They should revise.

Authors’ response: Dear Reviewer, Thank you so much; we revised it

------ Done

Authors’ response#19: Dear Reviewer, Thank you so much for your time. 

20 – Abstract, “The objective of this study was to assess the prevalence of and factors associated with uterine prolapse in women of reproductive age in Ethiopia.”: The objective was not accomplished by this study. UP can be asymptomatic. So, without a medical evaluation, it is not possible to determine the prevalence of UP in their population. The inclusion criteria were done by non-medical interview (using a questionary). The authors should revise it.

21 – Abstract, results, “The prevalence of symptomatic and anatomical uterine prolapse was 6.6% (28) and 5.9% (25), respectively”: Sorry, but the authors already said that 3 patients of the 28 symptomatic did not confirm the hypothesis of UP after medical examination. They did not revise the manuscript as previously indicated in question 04.

22 – Introduction, “UP is the most common gynecological health problem contributing to maternal morbidity and mortality in women of reproductive age in developing countries”: The authors should show the reference of their statement.

23 – Introduction: It is still too long. They should shrink it.

Authors’ response #23: Dear Reviewer, Thank you so much for your time. We revised it.

24 - The authors should describe the full words in first passage abbreviations (i.e. COR);

Authors’ response#24: Dear Reviewer, Thank you so much for your time. We revised

25 – The authors indexed the previous and the new revision. It was a little bit confusing to the identification of the new one.

Authors’ response#25: Dear Reviewer, Thank you so much for your time.

---

## [Decision Letter · Decision Letter 3]

9 Nov 2021

PONE-D-20-40659R3

Uterine prolapse and associated factors among reproductive-age women in south-west Ethiopia : a community-based cross-sectional study

PLOS ONE

Dear Dr. badacho,

Thank you for submitting your manuscript to PLOS ONE. After careful consideration, we feel that it has merit but does not fully meet PLOS ONE’s publication criteria as it currently stands. Therefore, we invite you to submit a revised version of the manuscript that addresses the points raised during the review process.

We look forward to receiving your revised manuscript.

Kind regards,

Richard Kao Lee, M.D.

Academic Editor

PLOS ONE

Reviewers' comments:

Reviewer's Responses to Questions

**Comments to the Author**

1. If the authors have adequately addressed your comments raised in a previous round of review and you feel that this manuscript is now acceptable for publication, you may indicate that here to bypass the “Comments to the Author” section, enter your conflict of interest statement in the “Confidential to Editor” section, and submit your "Accept" recommendation.

Reviewer #1: All comments have been addressed

Reviewer #3: (No Response)

Reviewer #5: (No Response)

2. Is the manuscript technically sound, and do the data support the conclusions?

Reviewer #1: Yes

Reviewer #3: No

Reviewer #5: (No Response)

3. Has the statistical analysis been performed appropriately and rigorously? 

Reviewer #1: Yes

Reviewer #3: No

Reviewer #5: (No Response)

4. Have the authors made all data underlying the findings in their manuscript fully available?

Reviewer #1: Yes

Reviewer #3: Yes

Reviewer #5: No

5. Is the manuscript presented in an intelligible fashion and written in standard English?

Reviewer #1: Yes

Reviewer #3: Yes

Reviewer #5: Yes

6. Review Comments to the Author

Reviewer #1: Authors improved their paper that now can be accept

Also the paper already done revised for 3 time I think now is correct to accept it

Reviewer #3: About the revised article “Uterine prolapse and associated factors among reproductive age women in southwest Ethiopia”: The authors’ aim was to evaluate the prevalence and factors associated with uterine prolapse among women of reproductive age throw a cross-sectional study. The end point was to analyze risk factors associated with uterine prolapse.

Unfortunately, I already made many suggestions to enhance the article. I and the authors have disagreement about many topics. The last 2 revisions (included the last one) have not evolved.

Cordially

Reviewer #5: Line 99, if the sample size was estimated using confidence interval, please state the level of % confidence used for your calculation. E.g. The sample size was calculated to ensure that the two-sided 95% confidence interval (CI) for the estimated prevalence will be within +/- 0.05 by using a single population formula with a proportion of 0.5.

Line 104, “theseareas” should be “these areas”.

Lines 104-108, it’s not clear about the sample allocations among kebeles. Please add one table to list the numbers of households for each of the eight kebeles, and number of selected households and subjects for this study.

Lines 109, “households were selected by systematic random sampling method”. If a selected household didn’t have a woman or have multiple women meeting the inclusion criteria, how did you select the subjects?

Line 196, “p < 0.005” should be “p < 0.05”.

Lines 237-239, “Even though we determined the sample using the proportion of 0.5 to get the maximum sample size, the sample was underpowered because prior prevalence estimation was more than eight times higher than what we found”. This statement should be removed. It was not underpowered because the sample size was planned as the maximum to ensure the width of 95% CI of an estimated prevalence will be within 0.1.

7. PLOS authors have the option to publish the peer review history of their article (what does this mean?). If published, this will include your full peer review and any attached files.

Reviewer #1: No

Reviewer #3: No

Reviewer #5: No

---

## [Author Response · Author response to Decision Letter 3]

14 Dec 2021

PONE-D-20-40659R3

Uterine prolapse and associated factors among reproductive-age women in south-west Ethiopia: a community-based cross-sectional study

Subject: Point by point response to reviewers’ comments

 Dear PLOS ONE Editorial Manager,

Please kindly find the submitted revised manuscript with ID number PONE-D-20-40659R3 entitled “Uterine prolapse and associated factors among reproductive-age women in south-west Ethiopia: a community-based cross-sectional study” point to point response to the reviewer’s comment for kindly consideration. 

With kind regards,

Abebe Sorsa (Corresponding Author)

Author’s point by point response to Reviewer’s Responses to Questions

Reviewer’s Responses to Questions

Comments to the Authors

1. If the authors have adequately addressed your comments raised in a previous round of review and you feel that this manuscript is now acceptable for publication, you may indicate that here to bypass the “Comments to the Author” section, enter your conflict of interest statement in the “Confidential to Editor” section, and submit your "Accept" recommendation.

Reviewer #1: All comments have been addressed

Reviewer #3: (No Response)

Reviewer #5: (No Response)

Reviewer #1: All comments have been addressed

Authors’ response: Dear Reviewer, Thank you so much

Reviewer #3: (No Response)

Authors’ response: Dear Reviewer, Thank you so much

Reviewer #5: (No Response)

Authors’ response: Dear Reviewer, Thank you so much

2. Is the manuscript technically sounds, and do the data support the conclusions?

Reviewer #1: Yes

Authors’ response: Dear Reviewer, Thank you so much

Reviewer #3: No

Authors’ response: Dear Reviewer, Thank you so much. We have revised the manuscript 

Reviewer #5: No response

Authors’ response: Dear Reviewer, Thank you so much. We have revised the manuscript 

3. Has the statistical analysis been performed appropriately and rigorously?

Reviewer #1: Yes

Authors’ response: Dear Reviewer, Thank you so much

Reviewer #3: No

Authors’ response: Dear Reviewer, Thank you so much; we have revised the manuscript 

Reviewer #5: No response

Authors’ response: Dear Reviewer, Thank you so much. We have revised the manuscript 

4. Have the authors made all data underlying the findings in their manuscript fully available?

Reviewer #1: Yes

Authors’ response: Dear Reviewer, Thank you so much

Reviewer #3: Yes

Authors’ response: Dear Reviewer, Thank you so much 

Reviewer #5: No response

Authors’ response: Dear Reviewer, Thank you so much. We have revised the manuscript 

5. Is the manuscript presented in an intelligible fashion and written in Standard English?

Reviewer #1: Yes

Authors’ response: Dear Reviewer, Thank you so much,

Reviewer #3: yes 

Authors’ response: Dear Reviewer, Thank you so much 

Reviewer #5: Yes

Authors’ response: Dear Reviewer, Thank you so much,

6. Review Comments to the author

Reviewer #1: In my opinion can be accepted, the paper is very good and also the time related to peer review is too much, and I think to make a decision in a short time, no one year.

Authors’ response: Dear Reviewer, Thank you so much for your time and reviewing this manuscript

Reviewer #1: Authors improved their paper that now can be accept

Also the paper already done revised for 3 time I think now is correct to accept it

Authors’ response#1: Dear Reviewer, Thank you so much for your time.

Reviewer #3: About the revised article “Uterine prolapse and associated factors among reproductive age women in southwest Ethiopia”: The authors’ aim was to evaluate the prevalence and factors associated with uterine prolapse among women of reproductive age throw a cross-sectional study. The end point was to analyze risk factors associated with uterine prolapse.

Unfortunately, I already made many suggestions to enhance the article. I and the authors have disagreement about many topics. The last 2 revisions (included the last one) have not evolved.

Cordially

Authors’ response#1: Dear Reviewer, Thank you so much for your time. We tried to address all the comments in revisions. Please you can refer our previous revision response Reviewer #3: 

Reviewer #5: Line 99, if the sample size was estimated using confidence interval, please state the level of % confidence used for your calculation. E.g. The sample size was calculated to ensure that the two-sided 95% confidence interval (CI) for the estimated prevalence will be within +/- 0.05 by using a single population formula with a proportion of 0.5.

Authors’ response#1: Dear Reviewer, Thank you so much for your time, we have revised the manuscript 

Line 104, “these areas” should be “these areas”.

Authors’ response#2: Dear Reviewer, Thank you so much , corrected 

Lines 104-108, it’s not clear about the sample allocations among kebeles. Please add one table to list the numbers of households for each of the eight kebeles, and number of selected households and subjects for this study.

Authors’ response#3: Dear Reviewer, Thank you so much, we revised and added a table. Number of married reproductive age group women in each kebele. Please see lines 111 - 115.

Lines 109, “households were selected by systematic random sampling method”. If a selected household didn’t have a woman or have multiple women meeting the inclusion criteria, how did you select the subjects?

Authors’ response#4: Dear Reviewer, Thank you so much, we have revised. Out of an estimated 422 participants, the sample size was adjusted proportionally for the selected eight kebeles. Then, married women of reproductive age were selected by random sampling method, using a list of married reproductive age group from a registration book of health post as sampling frame provided by Health Extension Workers (HEWs) working at Health Post in each respective

Line 196, “p < 0.005” should be “p < 0.05”.

Authors’ response#4: Dear Reviewer, Thank you so much, corrected please line 201. 

Lines 237-239, “Even though we determined the sample using the proportion of 0.5 to get the maximum sample size, the sample was underpowered because prior prevalence estimation was more than eight times higher than what we found”. This statement should be removed. It was not underpowered because the sample size was planned as the maximum to ensure the width of 95% CI of an estimated prevalence will be within 0.1.

Authors’ response#5: Dear Reviewer, Thank you so much, we removed the statement

---

## [Decision Letter · Decision Letter 4]

17 Dec 2021

Uterine prolapse and associated factors among reproductive-age women in south-west Ethiopia : a community-based cross-sectional study

PONE-D-20-40659R4

Dear Dr. Badacho:

We’re pleased to inform you that your manuscript has been judged scientifically suitable for publication and will be formally accepted for publication once it meets all outstanding technical requirements.

Kind regards,

Richard Kao Lee, M.D.

Academic Editor

PLOS ONE

Additional Editor Comments (optional):

Reviewers' comments:

Reviewer's Responses to Questions

**Comments to the Author**

1. If the authors have adequately addressed your comments raised in a previous round of review and you feel that this manuscript is now acceptable for publication, you may indicate that here to bypass the “Comments to the Author” section, enter your conflict of interest statement in the “Confidential to Editor” section, and submit your "Accept" recommendation.

Reviewer #1: All comments have been addressed

Reviewer #5: All comments have been addressed

2. Is the manuscript technically sound, and do the data support the conclusions?

Reviewer #1: Yes

Reviewer #5: (No Response)

3. Has the statistical analysis been performed appropriately and rigorously? 

Reviewer #1: Yes

Reviewer #5: (No Response)

4. Have the authors made all data underlying the findings in their manuscript fully available?

Reviewer #1: Yes

Reviewer #5: (No Response)

5. Is the manuscript presented in an intelligible fashion and written in standard English?

Reviewer #1: Yes

Reviewer #5: (No Response)

6. Review Comments to the Author

Reviewer #1: (No Response)

Reviewer #5: (No Response)

7. PLOS authors have the option to publish the peer review history of their article (what does this mean?). If published, this will include your full peer review and any attached files.

Reviewer #1: **Yes: **Francesco Di Gennaro

Reviewer #5: No

---

## [Editor Report · Acceptance letter]

29 Dec 2021

PONE-D-20-40659R4 

Uterine prolapse and associated factors among reproductive-age women in south-west Ethiopia: a community-based cross-sectional study 

Dear Dr. Badacho:

I'm pleased to inform you that your manuscript has been deemed suitable for publication in PLOS ONE. Congratulations! Your manuscript is now with our production department. 

Kind regards, 

on behalf of

Dr. Richard Kao Lee 

Academic Editor

PLOS ONE